# The structural role of SARS-CoV-2 genetic background in the emergence and success of spike mutations: The case of the spike A222V mutation

**Tiziana Ginex**[1◉], **Clara Marco-Marín**[2,3◉], **Miłosz Wieczór**[4◉], **Carlos P. Mata**[5,6◉], **James Krieger**[5◉], **Paula Ruiz-Rodriguez**[7◉], **Maria Luisa López-Redondo**[2], **Clara Francés-Gómez**[7], **Roberto Melero**[5], **Carlos Óscar Sánchez-Sorzano**[5], **Marta Martínez**[5], **Nadine Gougeard**[2,3], **Alicia Forcada-Nadal**[2,3], **Sara Zamora-Caballero**[2], **Roberto Gozalbo-Rovira**[2], **Carla Sanz-Frasquet**[2], **Rocío Arranz**[5], **Jeronimo Bravo**[2], **Vicente Rubio**[2,3], **Alberto Marina**[2,3], **The IBV-Covid19-Pipeline**[2], **Ron Geller**[7], **Iñaki Comas**[2,8], **Carmen Gil**[1], **Mireia Coscolla**[7], **Modesto Orozco**[3,9], **José Luis Llácer**[2,3], **Jose-Maria Carazo**[5]*

**1** Centro de Investigaciones Biológicas Margarita Salas (CIB-CSIC), Madrid, Spain, **2** Instituto de Biomedicina de Valencia (IBV-CSIC), Valencia, Spain, **3** Centro de Investigación Biomédica en Red en Enfermedades Raras (CIBERER), Madrid, Spain, **4** Molecular Modeling and Bioinformatics, Institute for Research in Biomedicine (IRB Barcelona), The Barcelona Institute of Science and Technology (BIST), Barcelona, Spain, **5** Centro Nacional de Biotecnología (CNB-CSIC), Madrid, Spain, **6** Centro Nacional de Microbiología (CNM-ISCIII), Instituto de Salud Carlos III, Madrid, Spain, **7** I²SysBio, University of Valencia-CSIC, FISABIO Joint Research Unit Infection and Public Health, Valencia, Spain, **8** Centro para Investigación Biomédica en Red sobre Epidemiología y Salud Pública (CIBERESP), Valencia, Spain, **9** Department of Biochemistry and Biomedicine, University of Barcelona, Barcelona, Spain

◉ These authors contributed equally to this work.
* carazo@cnb.csic.es

## Abstract

The S:A222V point mutation, within the G clade, was characteristic of the 20E (EU1) SARS-CoV-2 variant identified in Spain in early summer 2020. This mutation has since reappeared in the Delta subvariant AY.4.2, raising questions about its specific effect on viral infection. We report combined serological, functional, structural and computational studies characterizing the impact of this mutation. Our results reveal that S:A222V promotes an increased RBD opening and slightly increases ACE2 binding as compared to the parent S:D614G clade. Finally, S:A222V does not reduce sera neutralization capacity, suggesting it does not affect vaccine effectiveness.

## Author summary

Since early 2020, the trajectory of the COVID-19 pandemic has mostly been shaped by the appearance of novel variants of the SARS-CoV-2 virus. Accordingly, much of the scientific effort has been directed toward the question of explaining, understanding, and predicting the evolutionary fate of individual mutations in the viral genome. In this article, we focus on A222V, a particular mutation in the Spike protein that emerged in Spain in mid-2020

**Data Availability Statement:** All the analysed 4,993,996 SARS-CoV-2 genomes (collected from December 2019 to April 2022) were retrieved from

the GISAID platform (https://www.gisaid.org). The atomic coordinates for [S:A222V + S:D614G]-1-up and S:D614G-1-up were deposited in the Protein Data Bank with codes 7QDG and 7QDH, respectively. The [S:A222V + S:D614G]-1-up, [S:A222V + S:D614G]-2-up, [S:A222V + S:D614G]-3-down, S:D614G-1-up and S:D614G-2-up cryo-EM density maps were deposited in the EM Data Bank with codes EMD-13916, EMD-13917, EMD-13918, EMD-13919 and EMD-13920, respectively. The motion corrected micrographs for [S:A222V + S:D614G] were deposited to EMPIAR. Aligned MD trajectories for all the glycan-free and glycosylated S:D614G and [S:A222V + S:D614G] mutants simulated in this study are available at: bioexcel-cv19-dev.bsc.es/#/browse?search=Spike%20mutation%20A222V.

**Funding:** In this research work, PRR and TG were supported by salaries from the European Commission–NextGenerationEU through the CSIC Global Health Platform established by EU Council Regulation 2020/2094. The authors would like to acknowledge economic support from: (a) the Spanish Ministry of Science and Innovation through grants: SEV-2017-0712 funded by MCIN/AEI/10.13039/501100011033 (CPM, JK, RM, COSS, MM, RA, JMC); 116880GB-I00 (JLL) and 120322RB-C21 (VR) funded within PID 2020; PID2019-104757RB-I00 funded by MCIN/AEI/10.13039/501100011033/ and "ERDF A way of making Europe", by the "European Union" (JMC, COSS, RM, MM, CPM) and 104477RB-100 (IC) funded within PID 2019; RYC-2015-18213 (salary of MC) and RYC-2015-17517 (salary of RG) funded by the Ramon y Cajal fellowship program; RTI2018-094399-A-I00 (JMC, COSS, RM, CPM, MC) funded within the Retos Investigación program; the CRIOMECORR project funded as ESFRI-2019-01-CSIC-16; (b) the Horizon 2020 program through Marie Skłodowska-Curie Individual Fellowships EnLaCES (Proposal 101024130, salary of JK) and EPIDNA (Grant ID: 894498, salary of MW); (c) the European Research Council (ERC) through grants: ERC-2018-SyG-810057 HighResCells funded within the EXCELLENT SCIENCE program (JMC, MM, COSS, salary of RM), Grant 871037 iNEXT DISCOVERY (JMC, MM, COSS) and grant CoG-101001038 funded within the Consolidator Grants program (IC); (d) Generalitat Valenciana through grants: SEJI/2019/011 (MC) and SEJI/2019/030 (JLL); (e) Comunidad de Madrid through grant: S2017/BMD-3817 (JMC); (f) Santander Bank through grant: CSIC-COVID19-082 (AM, VR, JB, JLL, RG) funded through the Fondo Supera Covid-19; (g) Instituto de Salud Carlos III through grants: COV20/00437 (AM, VR, JB, JLL) and COV20/00140 (IC), both

and reappeared independently in the AY.4.2 subvariant of Delta one year later. As reemerging mutations often indicate an evolutionary advantage, we explored potential mechanisms linking A222V to biologically relevant outcomes. Using serological, functional, structural, and computational approaches, we identified key molecular-level differences conferred by A222V that potentially explain its repeated emergence in different genetic backgrounds. Our results point to subtle changes in the dynamic behavior of the receptor-binding domain in the binding-competent "up" conformation, ones that affect receptor binding itself, but can also act synergistically with other mutations by changing the accessibility of key residues involved in molecular recognition.

## Introduction

Since the start of the COVID19 pandemic in early 2020 the causative agent, SARS-CoV-2, has been diversifying. Hundreds of variants have been identified, associated with thousands of mutations. Most of these mutations have no assigned role and they thrive in the population due to different stochastic population genetics forces such as founder or superspreading events [1]. However, a handful of variants are known to have evolved increased transmissibility and/or ability to reduce antibody responses. Most of those variants are characterized by a constellation of mutations, from five to more than 15, in the spike (S) protein. While some mutations have been deeply characterized, the role of most of these mutations in underlying the phenotypes associated with the variants and their mechanisms of action remain unknown. Furthermore, many of the mutations found in the most successful variants, including Alpha, Delta or Beta, have been seen before in other genetic backgrounds, usually in isolation, where they have not been able to thrive in the population.

The repeated emergence of the same mutations across time and lineages suggests the action of positive selection and a functional role for the less-characterized mutations. Nevertheless, the reasons underlying the genetic success of these mutations and the role of the genetic background remains to be defined. A spike mutation common to almost all the strains circulating today is S:D614G. This mutation appeared early in the pandemic and has been associated with higher fitness in experimental work *in vitro*, *in vivo* and in epidemiological settings [2–4]. In particular, S:D614G has been reported to affect the conformational landscape of the spike glycoprotein [5–8], so that a particular domain known as the Receptor Binding Domain (RBD), discussed further in this work, may be more exposed for increased interactions with the main cellular receptor ACE2 [9]. Consequently, S:D614G variants outcompeted the initial 614D Wuhan-Hu-1 variant across the globe, although the S:D614G mutation alone cannot explain the success of the associated variants and probably there was a combined effect with a founder event [1]. Likewise, a new variant emerged during the summer of 2020, 20E/EU1, which was characterized by the additional spike mutation, S:A222V. This mutation has been seen both before and after summer 2020 in other genetic backgrounds, including Delta (as indicated in the **Results** section), suggesting that the mutation had a functional effect and may explain the success of the associated variant. However, in-depth analysis of the population dynamics of the variant in different countries suggested that the success of 20E/EU1 was a by-product of lifting restrictions in Europe after the first lockdown and holiday-associated travels across Europe [10]. This opens the question of why S:A222V has been selected multiple times in the evolution of SARS-CoV-2 and what the functional role of the mutation in viral replication or transmission is.

funded through the Fondos Covid-19 initiative; (h) Spanish National Research Council through grant: 202080E110 funded within the PTI Salud Global initiative (AM, VR, JB, JLL, RG). The funders had no role in study design, data collection and analysis, decision to publish, or preparation of the manuscript.

**Competing interests:** NO authors have competing interests.

Here, we use S:A222V as an example of those mutations that have arisen multiple times in the SARS-CoV-2 spike protein in different genetic backgrounds but had no associated increased epidemiological fitness. Tracing the interactions with the genetic background in which they appear (epistasis) could help in predicting the potential success of these new mutations in the population. To clarify the role of S:A222V, we have used an extensive genomic dataset describing their population dynamics across SARS-CoV-2 lineages. We have also studied the role of the genetic background by combining the mutation S:A222V with S:D614G in neutralization experiments with pseudotyped viruses, followed by biophysical characterization of the mutant spike. On the structural side, we report here the cryo Electron Microscopy (cryo-EM) map and structural model of the 1-up conformation of 20E/EU1 (7QDG, EMD-13916) and the ubiquitous S:D614G mutant (7QDH, EMD-13919). Cryo-EM and Molecular Dynamics (MD) simulations were finally combined to obtain indications about the functional role of the mutation. This multidisciplinary approach allowed us to provide a plausible answer to the repeated emergence of the S:A222V mutations, which could be summarized as an allosteric effect resulting in an enhanced flexibility of the RBD in the up state, although its translation into significant phenotypic effects depends on the genetic background. Finally, serological neutralization studies and *in vitro* thermostability assays did not show a marked difference between the two strains, suggesting it does not affect vaccine effectiveness.

## Results

### Population dynamics of S:A222V through space and time

The mutation S:A222V first arose within Lineage B.1.177, described for the first time in Spain in Summer 2020 [10]. S:A222V has been observed in 447,777 sequences from 123 countries by April 2022, almost exclusively co-occurring with S:D614G (99.86%). Additionally, 9% of 4,993,996 SARS-CoV-2 sequences available until 2022-04-01 include [S:A222V + S:D614G] and only 0.012% harbour S:A222V without S:D614G, indicating that the presence of S:A222V is tightly linked to S:D614G. Interestingly, although S:A222V arose with B.1.177, by September 2021 more than half of the sequences with S:A222V were not in the genomic context of B.1.177 (**Table A in S1 Text** and **Fig 1A**), indicating that although B.1.177 was replaced by other variants [10], S:A222V emerged independently in different lineages (**Fig 1A**). Non-B.1.177 sequences with S:A222V were isolated in 141 countries and classified in as many as 364 PANGO lineages (**Table A in S1 Text**), which indicates subsequent and independent appearances of S:A222V in different genetic backgrounds (**Fig 1A**).

By July 2021, the majority of the non-B.1.177 occurrences appeared in lineage B.1.617.2, where 10.35% of B.1.617.2 contain S:A222V (**Fig 1B**). B.1.617.2 is also known as Variant Of Concern (VOC) Delta and it includes all AY lineages [11] (see below and Fig A in S1 Text). Less frequently, S:A222V appears in 0.13% of B.1.1.7 (VOC Alpha), and 0.56% of B.1 overall. The temporal pattern for S:A222V (blue area in **Fig 1B**) shows two peaks, the first corresponding to the dynamics of B.1.177 (green line in **Fig 1B**) and the second to the expansion of B.1.617.2 (VOC Delta) (red line in **Fig 1B**). We did not observe a similar peak in S:A222V during the increase of B.1.1.7 (VOC Alpha; orange line in **Fig 1B**), possibly suggesting that epistatic interactions may favour the transmission of S:A222V preferentially in some genetic backgrounds. Sequences with S:A222V began to decrease with the replacement of VOC Delta (B.1.617.2; red line in Fig 1B) by VOC Omicron [12] (B.1.1.529; blue line in Fig 1B). By April 2022, S:A222V only appears in 0.05% of 152,891 sequences classified as B.1.1.529, and due to the predominance of B.1.1.529, S:A222V is almost no detectable (blue area in Fig 1B).

Another seven lineages harbour S:A222V in high or medium frequencies, although those lineages are very rare with very few sequences and are very restricted geographically. Out of

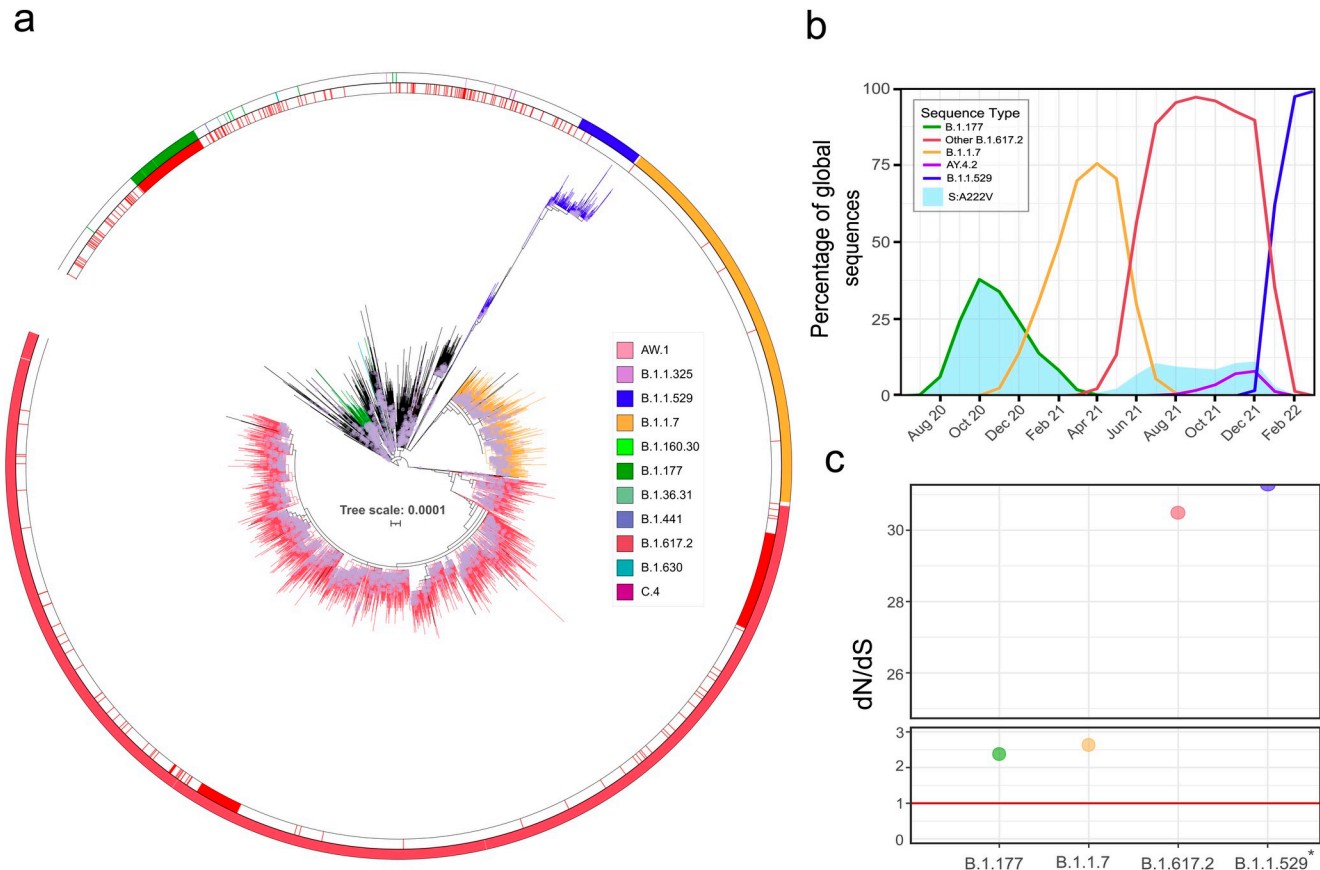

**Fig 1. Temporal and phylogenetic distribution of S:A222V.** (a) Global phylogeny of 11,166 sequences belonging to G clade. The red inner circle denotes sequences with S:A222V, the external circle indicates PANGO linages of interest indicated in legend. (b) Percentage of global sequences with S:A222V (blue area) in different VOCs. Sequences belonging to parental lineages designated as B.1.177 appear as a green line, B.1.1.7 as a yellow line, B.1.617.2 as a red line, AY.4.2 as a purple line, and B.1.1.529 as a blue line. (c) The ratio of nonsynonymous to synonymous substitution rates (dN/dS) for the codon 222 of the spike for each period with a predominant variant, the size of the circle indicates–log(p) with a constant value of 10. *dN/dS for B.1.1.529 period is even higher but cannot be represented because dS = 0.

397 sequences classified as B.1.36.31, 93.45% harbour S:A222V (Fig B in S1 Text). Similarly, 91.66% of 12 sequences classified as B.1.630 harbour S:A222V (Fig B in S1 Text). Other rare lineages such as B.1.1.325, C.4, AW.1, B.1.1.160.30, and B.1.441 include a percentage of sequences with S:A222V that ranges from 49% to 15% (Fig B in S1 Text, Table A in S1 Text).

To decipher the recent dynamics of S:A222V, we focused on B.1.617.2 in a global dataset of 2,992,547 sequences (**Fig 2A** and **2B**). B.1.617.2 sequences with S:A222V belong to different sub-lineages, being characteristic of five sub-lineages: AY.4.2, AY.9, AY.26, AY.27, and AY.47, where it is present in more than 95% of the sequences of each sub-lineage (**Fig 2B**). However, S:A222V is also present in 25% of sequences classified as the parental B.1.617.2 lineage, and in 1.31% of other B.1.617.2 sub-lineages (**Fig 2B**). S:A222V has appeared at least 32 times in the genomic context of B.1.617.2 (**Fig 1A, Fig B in S1 Text**), being detected in at least 171 different sub-lineages (**Fig 2B and Fig B in S1 Text**).

S:A222V appeared within B.1.617.2 as early as March 2021, when it accounted for 17% of all B.1.617.2 sequences (**Fig 2A**). After that, B.1.617.2 sequences with S:A222V increased to 23% in April 2021 but dropped afterwards. This decrease is mainly driven by the decrease of the sequences designated as parental lineage B.1.617.2 (**Fig 2B** and **Fig A in S1 Text**). Conversely, since July 2021, the proportion of B.1.617.2 (and S:A222V along with it) has been

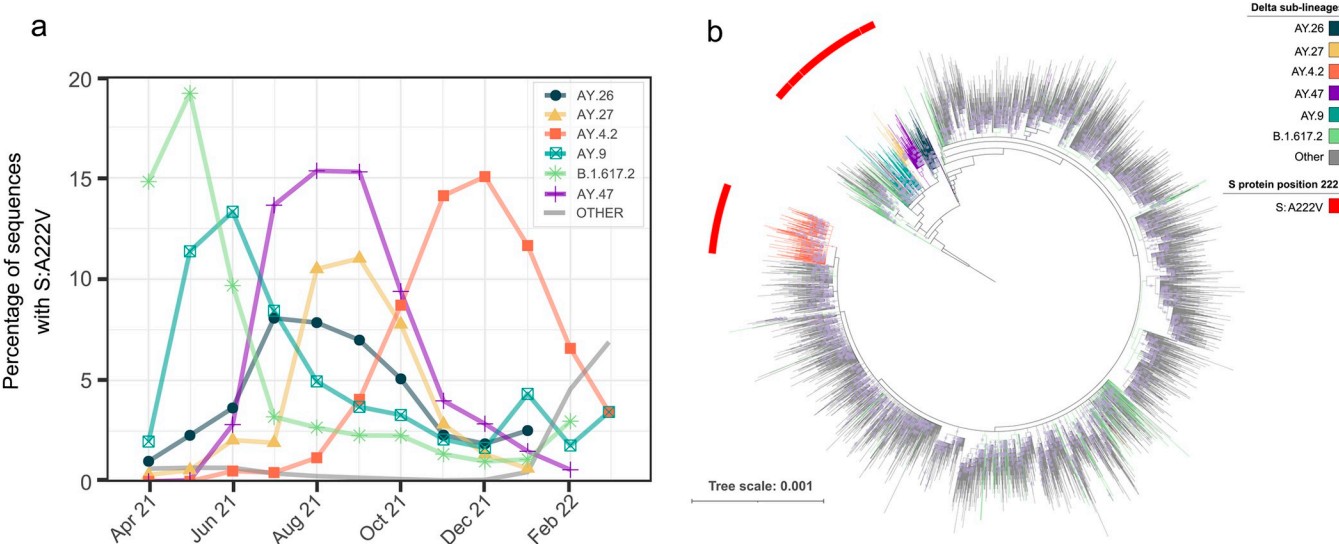

**Fig 2. Temporal and phylogenetic distribution of the S:A222V in B.1.617.2 sequences.** (**a**) Percentage of sequences designated as parental B.1.617.2 and its sub-lineages (designated as AY) with S:A222V by month. (**b**) Maximum likelihood phylogeny of 10,049 sequences from Lineages B.1.617.2 and derivatives. Phylogeny rooted with reference MN908947.3. Red external ring indicates sequences with S:A222V, the colour of branches indicates B.1.617.2 sub-lineages of interest. The scale bar indicates the number of nucleotide substitutions per site. Each circle in branches correlates with the bootstrap value; only bootstraps from 70 to 100 are represented.

increasing mainly due to the increase of AY.47 and AY.4.2 (**Fig 2A**). Among the different appearances of S:A222V within B.1.617.2, it has been successfully transmitted on at least two occasions: one in the last common ancestor of AY.4.2 and the other in the common ancestor of Lineages AY.9, AY.26, AY.27 and AY.47 (**Fig 2B**). Additionally, eight other less abundant lineages which derive from the same common ancestor as AY.9, A.26, AY.27 and AY.47 also have S:A222V (**Fig 2B** and **Fig B** in **S1 Text**).

In order to assess the evolutionary pressures on S:A222V, we employed the phylogenetically corrected single-likelihood ancestor counting (SLAC) method [13] to calculate the ratio of nonsynonymous to synonymous substitution rates (dN/dS) estimates for 874 codons in the NTD (N-Terminal Domain) region of the spike. dN/dS can potentially estimate the strength and direction of selection, typically indicating positive selection when dN/dS> 1. Using three thousand haplotypes representative of G-clade, we confirmed previous results [14] that S:A222V shows signatures of positive selection, with an estimate of dN/dS of 1.63 (p < 0.0001). This result was consistent with the analysis of a wider dataset of eleven thousand haplotypes (dN/dS of 1.62; p < 0.0001). When analysing independently four main periods of the pandemic (see Methods), we could detect a temporal pattern in point estimates of dN/dS, with values higher than one during all periods analysed (Fig 1C). dN/dS showed values of 2.38 and 2.64 when B.1.177 and B.1.1.7 dominated, and a value of 30 when B.1.617.2 appeared, and even a higher dN/dS value for B.1.1.529 period due to dS = 0 (Fig 1C), indicating the presence of signals of positive selection in S:A222V during different waves, but especially after B.1.617.2 arose.

## Neutralization of pseudotyped VSV by convalescent sera

One possibility for the selection of S:A222V could be an improved ability to replicate in immune populations. As vaccination had not yet started in Spain when this mutation was first observed in 2020, any such effect would have to result from selection for escape from existing

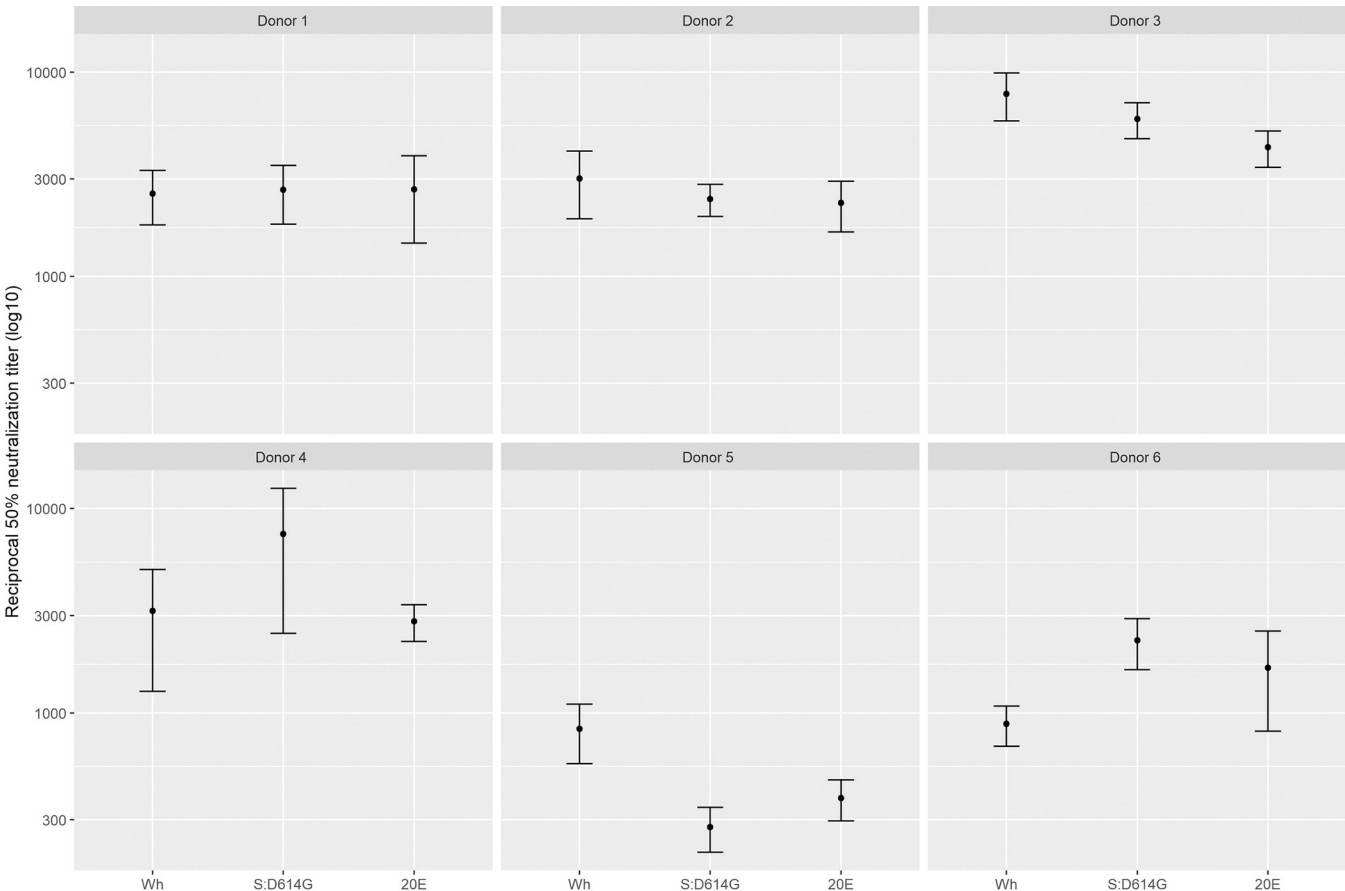

**Fig 3. S:A222V does not alter the neutralization capacity of convalescent sera.** The sensitivity to neutralization by convalescent sera of pseudotyped VSV carrying the ancestral Wuhan-Hu-1 (Wuhan), S:D614G, or the [S:A222V + S:D614G] (20E) spike protein was evaluated. No significant differences were detected between any of the sera (n = 6; p>0.05 by Kruskal-Wallis test; p>0.05 by Mann-Whitney test for Wuhan-Hu-1 vs. S:D614G, Wuhan-Hu-1 vs. 20E, or S:D614G vs. Wuhan-Hu-1).

immunity in convalescent individuals. To test this possibility, we evaluated the ability of VSV pseudotyped with either the ancestral Wuhan-Hu-1 spike protein, S:D614G, or 20E ([S:A222V + S:D614G]) to be neutralized by sera from the first wave of infection in Spain (April 2020) when no S:A222V mutations were circulating (**Fig 1B**). No significant differences were observed in neutralization between the different viruses (**Fig 3**).

### *In vitro* functional assays

The ectodomain of SARS-CoV-2 protein S corresponding to the Wuhan-Hu-1, S:D614G or [S:A222V + S:D614G] variants, hosting proline substitutions at residues 986 and 987, a mutant non-cleavable sequence at the furin site and a C-terminal foldon trimerization motif, were flash-purified (1 day) identically and used fresh. The three forms were indistinguishable by SDS-PAGE and eluted identically from a SEC column (shown for S:D614G and [S:A222V + S:D614G] spikes in **Fig C a-b in S1 Text**), as expected if they had essentially the same size and glycosylation pattern. Their quality, assessed by negative-stain electron microscopy (EM), was also similarly high (shown for S:D614G in **Fig C c in S1 Text**).

To functionally characterize the purified protein S variants, we first used thermal shift assays to measure their thermostability and we found that [S:A222V + S:D614G] and S:D614G

**Table 1.** *In vitro* functional assays.

| Protein | Spike-ACE2 binding by Biolayer Interferometry[a] | | | Thermal shift[a] |
|---|---|---|---|---|
| | $K_D$ (M) | $k_{on}$ (1/M.s) | $k_{off}$ (1/s) | $T_{0.5}$ (˚C) |
| Wuhan-Hu-1 | 7.93E-8 ± 0.99E-8 | 2.10E5 ± 0.26E5 | 1.62E-2 ± 0.09E-2 | 53.30 ± 0.00 |
| S:D614G | 6.57E-8 ± 0.74E-8 | 3.03E5 ± 0.38E5 | 2.02E-2 ± 0.11E-2 | 55.79 ± 0.01 |
| [S:A222V + S:D614G] | 4.96E-8 ± 1.23E-8 | 4.00E5 ± 0.65E5 | 1.82E-2 ± 0.09E-2 | 55.75 ± 0.07 |

[a] Results are given as mean ± SE

variants show very similar half-melting temperatures ($T_{0.5}$) and are both slightly more stable than the original Wuhan-Hu-1 variant (**Table 1** and **Fig D in S1 Text**). Then, to see the possible impact of the S:A222V mutation on the spike binding capacity to the protein ACE2, we carried out protein-protein interaction assays by biolayer interferometry (BLI) (**Table 1** and **Fig E in S1 Text**). The affinities for ACE2 were somewhat higher for the [S:A222V + S:D614G] and S:D614G variants than for the Wuhan-Hu-1 variant ($K_D$ values of 50 nM, 66 nM and 79 nM, respectively), in agreement with previous reports comparing the S:D614G and Wuhan-Hu-1 variants [9]. Although the dissociation constants for proteins carrying S:D614G and [S:A222V + S:D614G] mutations are similar, we observed a higher $k_{on}$ for the [S:A222V + S:D614G] mutant, perhaps reflecting structural differences related to higher accessibility of the RBD in this mutant compared to that in the S:D614G or the Wuhan-Hu-1 variants. To investigate this possibility further, we determined the cryo-EM structures of both the [S:A222V + S:D614G] and the S:D614G S proteins.

## Structural analysis of the allosteric role of S:A222V and epistatic interactions with S:D614G

To provide a comparative structural analysis of the S:A222V mutation on the background of S:D614G, we combined cryo-EM and classical MD simulations. Samples of spikes with both [S:A222V + S:D614G] and S:D614G, as an internal control, were produced for cryo-EM as described in **Methods**. Data about cryo-EM on the S:D614G mutant as well as a detailed description of the data sources, the modelled systems, including model validation metrics, the MD protocols and structural stability is reported in **Methods** and **S1 Text**.

**Cryo-EM of the [S:A222V + S:D614G] mutant.** Movies were collected as described in **Methods**. A representative micrograph is shown in **Fig F a in S1 Text**, while representative 2D class averages are shown in **Fig F b in S1 Text**. We followed both a "standard" Single Particle Analysis workflow, shown in **Fig G a in S1 Text**, and a modified one specifically aimed at detecting deviation from axial symmetry through symmetry relaxation, presented in **Fig G b in S1 Text**. Note that all 3D classifications have been performed multiple times [15] and that assignment into classes is provided with mean and standard deviation information.

The sample, under the conditions indicated in **Methods**, presented a majority conformation 1-up with about 50% of the particles (after consensus), although two minority but stable 2-up (1.6%) and 3-down (1.9%) conformations were also found through symmetry relaxation (**Fig G in S1 Text**). The highest resolution structure of the consensus 1-up type is shown in **Fig 4A**, together with quality Figs in **Fig 4B** such as the global FSC curve, the reported resolution (3.4 Å) and the angular coverage. A structural model of the 1-up conformation was derived from the map by model building and refinement procedures (**Methods** and **Fig 4C**). All structures have been submitted to the PDB and EMDB (for more details see **Data Availability**). As a control, a cryo electron microscopy analysis of the S:D614G mutant was also performed and

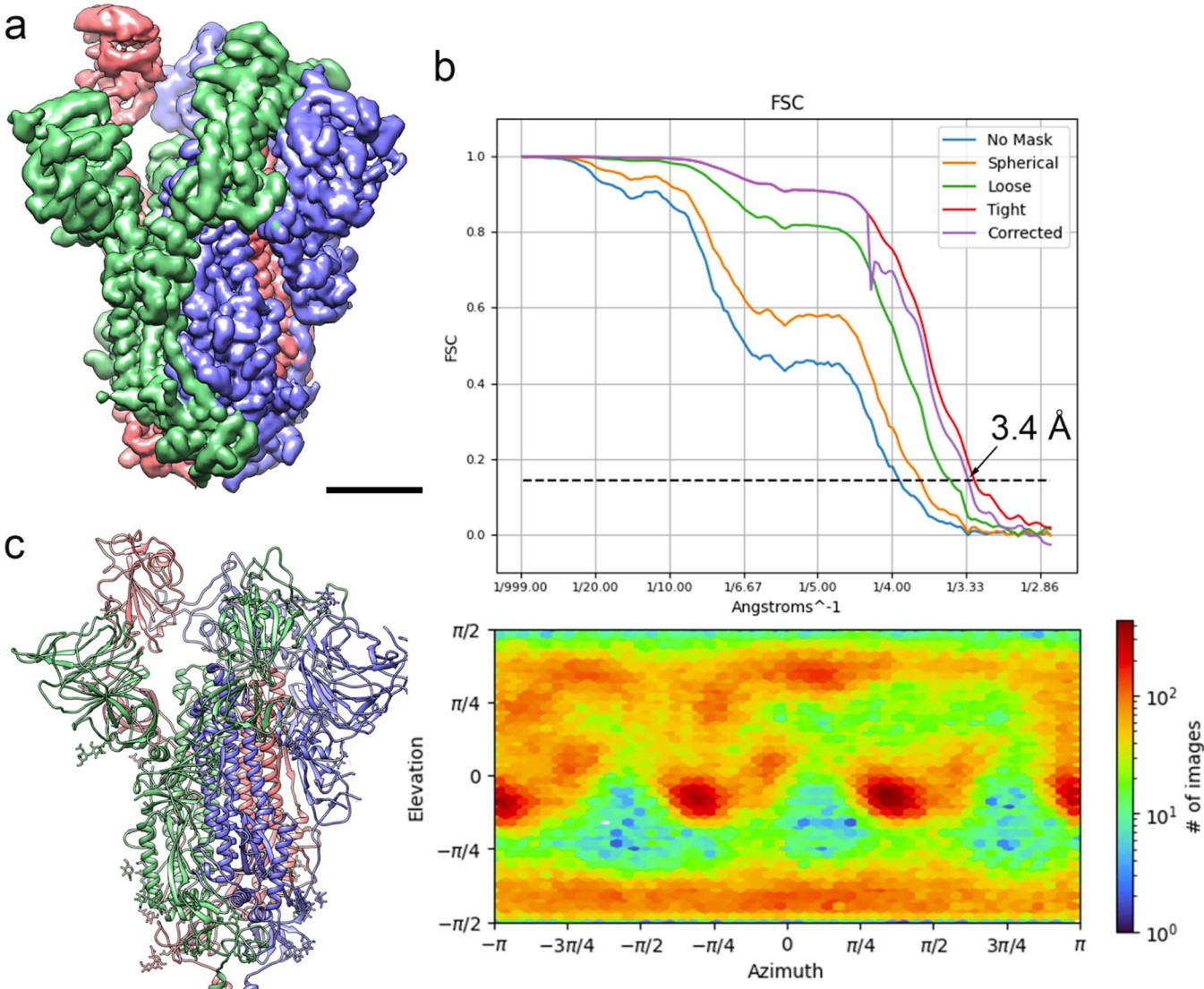

**Fig 4. Cryo-EM of the [S:A222V + S:D614G] mutant.** (**a**) Side view of the cryo-EM density map of the [S:A222V + S:D614G] mutant obtained after consensus of three independent runs of symmetry relaxation 3D classification, followed by initial model generation, refinement without symmetry imposition and deepEMhancer map sharpening, shown at 3 σ. Bar = 30 Å. (**b**) Fourier Shell Correlation (FSC) resolution curve (top) shown as the regular cryoSPARC global FSC resolution output, which includes no mask and different masks. Resolution based on the gold standard 0.143 criterion is 3.4 Å. The angular distribution coverage profile is also shown (bottom). (**c**) Atomic model of the [S:A222V + S:D614G] mutant shown as ribbon diagrams. S protein subunits are coloured in blue (chain A), red (chain B) and green (chain C). Glycan molecules are shown as stick diagrams and coloured according to their corresponding subunits.

presented in **Figs H and I in S1 Text.** A detailed summary of cryo-EM data acquisition parameters together with model validation metrics are reported in Table B in S1 Text.

**Structural comparison of the [S:A222V + S:D614G] and S:D614G 1-up mutants.** Comparison with the structure of the S:D614G mutant reveals that in the S:A222V mutation the V222 residue is accommodated in a highly hydrophobic environment of the NTD shaped by V36, Y38, F220 and I285 (**Fig 5A**). As observed in **Fig J in S1 Text,** this mutation is expected to slightly affect the interaction of the open subunit (chain B; $NTD_B$) with the neighbouring $CTD1_A$. Overall, the structures for the two mutants are strikingly similar (RMSD of 1.332 Å over 3193 Cα atoms), especially at the level of the S2 region, although we can observe small

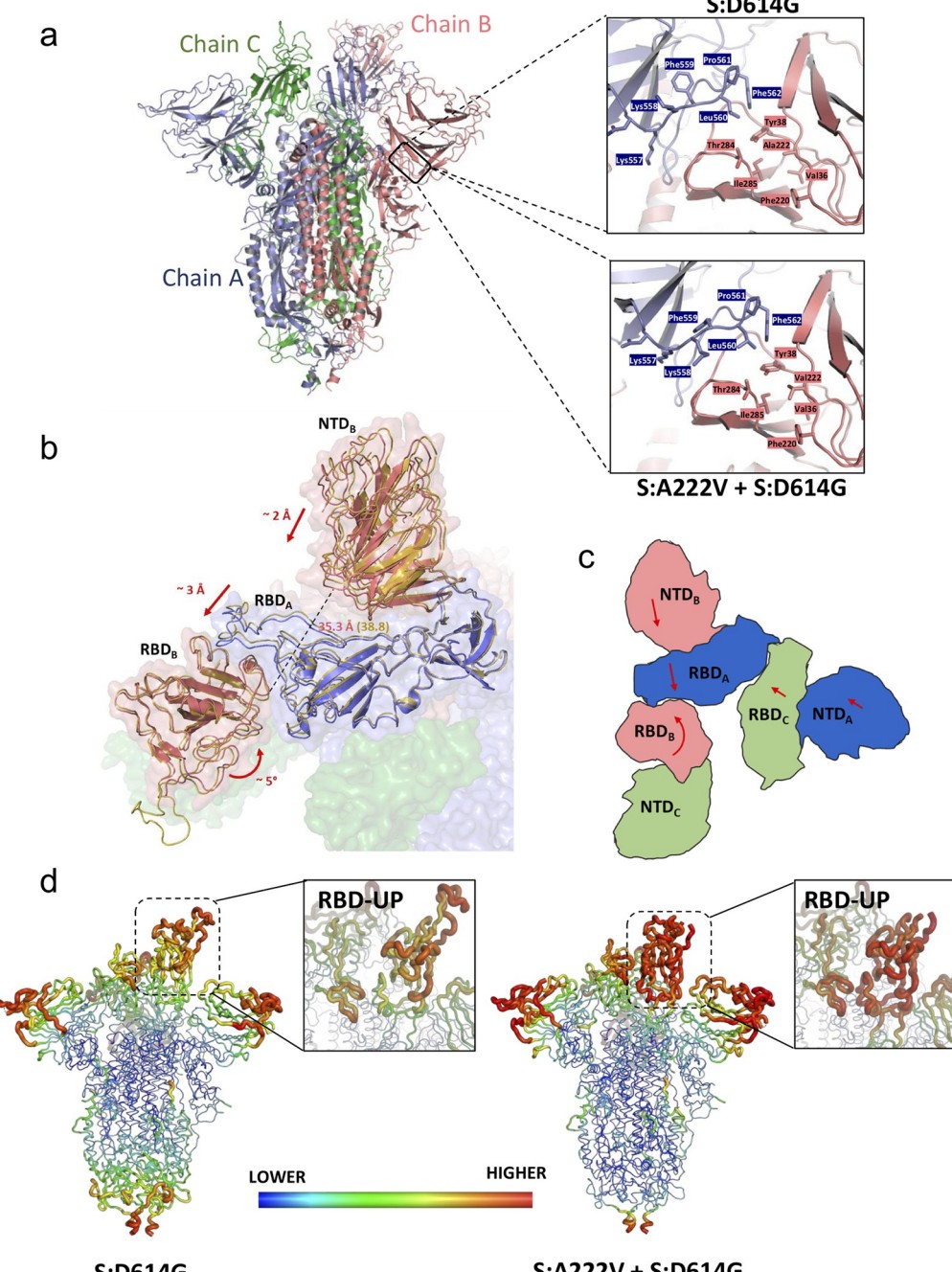

**Fig 5. Effects of the S:A222V mutation on the structure of SARS-CoV-2 spike.** (**a**) On the left, cartoon representation of the trimeric spike. On the right, detail of the region of interaction between the NTD from subunit B (salmon) and the CTD1 region from subunit A (blue). The side chains of residues surrounding A222 (upper panel) or V222 (lower panel) are shown as sticks. (**b**) Comparison of the structures for the S:D614G and [S:A222V + S:D614G] mutants showing main conformational changes observed when S:A222V is present. Detailed view of the trimeric spike surface shown in semi-transparent representation with different colours for the different subunits, and $RBD_B$, $RBD_A$ and $NTD_B$ domains from the S:D614G and [S:A222V + S:D614G] mutant structures shown as cartoon. Domains corresponding to the S:D614G structure are coloured in yellow. Arrows indicate the direction and magnitude of the observed domain movements. A dashed line represents the distance between the Cα atoms of residues 114 and 381 in chain B, which is ~ 3 Å higher in S:D614G than in [S:A222V + S:D614G]. (**c**) Schematic representation of the RBD and NTD from different subunits to show the movements observed when comparing the two structures (**d**) Comparison of the B-factor of $RBD_B$ of the two spike mutants. Backbone is coloured and sized according to the B-factor values for S:D614G (left) and [S:A222V + S:D614G] (right).

conformational changes in the position of the NTDs and RBDs. These differences are particularly relevant for the RBD in the up position (RBD$_B$), which is involved in the interaction with the host receptor, ACE2. Indeed, superimposition of these two structures with that of a spike in complex with bound ACE2 (PDB: 7DF4) indicates that in these two spike structures small rearrangements of this RBD-up would be beneficial to bind ACE2 [16]. The conformation of subunit C is quite similar in the two structures (RMSD 1.108 Å over 1074 Cα atoms), in contrast with the other two subunits (RMSD of 1.381 Å over 1070 Cα atoms for chain A and 1.389 Å over 1049 Cα atoms for chain B), which differ especially in (i) RBD$_B$, (ii) RBD$_A$, and (iii) NTD$_B$ (**Fig 5B** and **S1 Movie**). The analysis of these 3 domains allows us to describe the main conformational changes between these two spike structures. RBD$_A$ is sandwiched between RBD$_B$ and NTD$_B$. We observed that RBD$_A$ and NTD$_B$ move together as a rigid body (RMSD of these two domains between [S:A222V + S:D614G] and S:D614G structures is only of 0.726 Å), approaching RBD$_B$ in the case of [S:A222V + S:D614G] when compared to that in the S:D614G structure. On the other hand, RBD$_B$ of [S:A222V + S:D614G] undergoes a rotational movement that also brings it closer to RBD$_A$ (**Fig 5B** and **5C,** and **S1 Movie**). All these movements affect the degree of tightening of the S1 region of the spike (**S1 Movie**), so that the global conformation of the [S:A222V + S:D614G] mutant seems to be tighter than that of S:D614G. This is mainly a consequence of the tightening of subunit B around RBD$_A$. In this direction, the distance between residues 114 and 381 (**Fig 5B**), located in NTD$_B$ and RBD$_B$ at the interface between these domains and the RBD$_A$, are 3.5 Å closer in the [S:A222V + S:D614G] structure (35.3 Å in [S:A222V + S:D614G] vs. 38.8 Å in S:D614G). Although to a lesser extent, the distance between these residues in subunit A is also shorter in [S:A222V + S:D614G] when compared to that in S:D614G (42.4 Å in [S:A222V + S:D614G] vs. 43.9 Å in S:D614G). These tightening movements also increase the surface of contact between the NTD and RBD of the different subunits in [S:A222V + S:D614G] when compared to those in S:D614G (**Table C in S1 Text**). This effect seems to correlate with neither an increase in the stability of the S protein, as suggested by thermal shift stability assays (see above), nor with a decrease in the flexibility of the protein. In fact, an analysis of the temperature factors (B-factors) of the two structures (**Fig 5D**) shows a much higher mobility for RBD$_B$ in [S:A222V + S:D614G]. Not surprisingly, the regions in direct contact with RBD$_B$ (that is, RBD$_A$ and the contacting loops of NTD$_C$) also show increased B-factor values.

**Molecular dynamics simulations of the S:D614G and [S:A222V + S:D614G] mutants.** 300 ns MD simulations run in triplicate for each of the two simulated systems in the 3-down (DDD), closed state, yielded a total of 0.9 μs of sampling per system (DDD$_{S:D614G}$ and DDD$_{[S:A222V + S:D614G]}$ models in Table D in S1 Text). Given the metastable nature of the open, 1-up (UDD) conformation of the Spike protein, longer 500-ns simulations were run in this case, totalling 1.5 μs per system (UDD$_{S:D614G}$ and UDD$_{[S:A222V + S:D614G]}$ models in **Table D in S1 Text**). 200 ns of MD simulations were also run for both S:D614G and [S:A222V + S:D614G] mutants based on atomistic structures fitted to our cryo-EM densities (*cryo*-UDD$_{S:D614G}$ and *cryo*-UDD$_{[S:A222V + S:D614G]}$ models in **Table D in S1 Text**).

We initially analysed these simulations *via* principal component analysis (PCA) focusing on a single S1 subunit (**Methods**). The projections on the first two eigenvectors for the MD simulated mutants, collectively capturing 82% of the structural variance within the conformational landscape explored by the S1, are shown in **Fig 6A**. The largest motion represented by the first eigenvector (PC1: 72%) mainly describes the fluctuation of the RBD from the closed to the open state, with a minor contribution from the NTD of the same subunit (**Fig 6B**). The second motion (PC2: 10%) represents the lateral fluctuation of the RBD along with lateral movements of the NTD (**Fig 6C**), which affects the orientation of the solvent-exposed RBD region in the open state. As shown in **Fig 6A**, while the closed states of both mutants populate

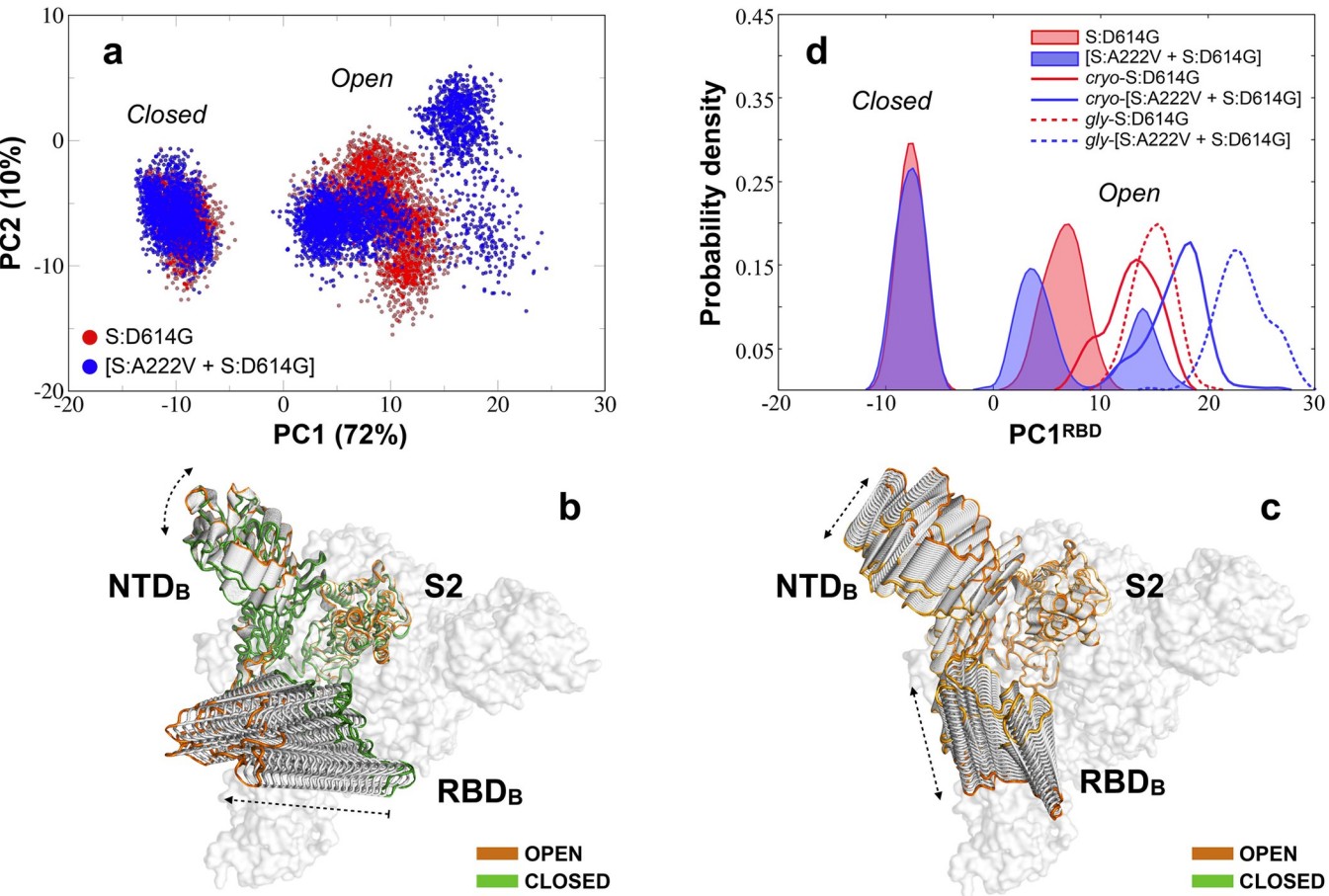

**Fig 6. PCA for the three MD simulated systems.** (**a**) PCA projection of the first two high-variance motions for the S:D614G (in red) and [S:A222V + S:D614G] (in blue) mutants of SARS-CoV-2 S1 subunit of the glycan-free spike protein. (**b**) Conformational evolution of the NTD and RBD along the first eigenvector (PC1) mainly accounting for the RBD *closed-to-open* transition. The open and closed states at the two ends of the PC1 are highlighted in orange and green, respectively. (**c**) Conformational evolution of the NTD and RBD along the second eigenvector (PC2) accounting for lateral RBD+NTD fluctuations. An orange gradient is applied here to distinguish between different lateral orientations of the open NTD-RBD pair at its two ends. (**d**) Continuous population densities along the PC1 for a PCA relative to the RBD (residues 330–530; PC1$^{RBD}$): solid profiles are used for the simulated closed, 3-down and open, 1-up states from the glycan-free systems based on 6VXX and 6VSB while continuous and dashed lines are used to define population densities for the simulated open 1-up states of the cryo-EM and fully-glycosylated S:D614G (red) and [S:A222V + S:D614G] (blue) mutants.

an overlapping well-defined region of the PCA space, the open state of the [S:A222V + S: D614G] mutant is observed to explore a visibly larger range of conformations in a clearly multimodal manner.

To more directly trace the *closed-to-open* transition of the RBD, we recalculated the PCA using RBD atoms (residues 330–530) and plotted continuous population densities along PC1 for the open and closed RBD domain of the 1-up and 3-down models of the two mutants (**Fig 6D**). As expected from **Fig 6A**, the open RBD of the [S:A222V + S:D614G] mutant sampled a bimodal density distribution (values centered on 4 and 14 arbitrary units for PC1$^{RBD}$), with the leftmost substate largely overlapping with the unimodal distribution of the S:D614G mutant (values centered on 8 for PC1$^{RBD}$; **Fig 6D**). Density distributions for the S:D614G and [S:A222V + S:D614G] mutants from the simulations of the 1-up cryo-EM structures are also reported in **Fig 6D** (unfilled solid lines). Comparison with the reference distributions (red and blue filled profiles in **Fig 6D**) revealed that both are slightly shifted to more positive values (respectively centred on 14 and 19) along the PC1$^{RBD}$. Let us remark that the cryo-EM [S:

A222V + S:D614G] mutant still explores a more open distribution with respect to the S:D614G mutant, thus confirming our finding. Furthermore, it seems to have no observable effect on the closed 3-down state.

To characterize the relative impact of glycosylation on the conformational preferences observed for the previously analysed glycan-free systems, we also simulated both mutants from the fully glycosylated structure (**Methods**). Since no relevant effects, relative to the observed phenomenon, were produced by the S:A222V mutation on the conformational ensemble of the 3-down closed state (see above), MD simulations were limited to the open 1-up state of the protein for this analysis. Accordingly, 400 ns of MD simulations were run for the S:D614G and [S:A222V + S:D614G] mutants (*gly*-UDD$_{S:D614G}$ and *gly*-UDD$_{[S:A222V + S:D614G]}$ models in **Table D in S1 Text**). As before, the *closed-to-open* transition of the RBD was analysed by performing PCA on the RBD (residues 330–530) and continuous density distributions for PC1$^{RBD}$ were finally plotted (red and blue dashed lines in **Fig 6D**). In line with other published data [17–19], glycosylation seems to affect the extent of the RBD opening. Even more shifted density distributions were in fact registered for both the S:D614G and [S:A222V + S:D614G] mutants (respectively centred on 15 and 22 for PC1$^{RBD}$) with respect to the previously simulated systems. Nevertheless, we still observe the stabilization of a more open conformation of the RBD for the glycosylated [S:A222V + S:D614G] mutant, thus confirming the general trend registered for our more extensive, microsecond-scale explorations of the glycan-free mutants (**Fig 6A** and **6D**).

To account for potential interdomain allosteric implications, we also analysed the effects induced by the RBD opening on the other two closed RBDs within the same trimeric ensemble of the simulated 1-up trajectories (**Fig K a, c, and e in S1 Text**). For completeness, data obtained from this analysis were also compared with those for the whole S1 units (**Fig K b, d, and f in S1 Text**). A slightly higher dynamical behaviour was observed at the level of the closed S1/RBD units for the simulated (glycan-free) cryo-EM mutants (**Fig K c and d in S1 Text**), with values spreading between -15 and 5 along the PC1$^{S1/RBD}$. A tendency to more negative values for the two closed subunits (blue vs. red profiles in **Fig K c and d in S1 Text**) of the [S:A222V + S:D614G] mutant was observed, especially for the glycan-free cryo-EM trajectories, thus suggesting the possibility of a certain uncoupling of its trimeric S1 (RBD-NTD) ensemble. We also performed a mutational free energy analysis on the open, 1-up structures (**Methods**). Results from this analysis (**Table E in S1 Text**) indicated that the impact of the S:A222V mutation on the RBD opening is too small to establish a significant preference for either of the conformational states of the RBD.

**Dynamic network analysis.** To identify direct mechanical connections responsible for a more open RBD conformation in the [S:A222V + S:D614G] mutant, we performed dynamic network analysis on an adjacent RBD-NTD pair (see **Fig 7A** and **7B**), i.e. with the two domains coming from two different chains.

The relative values of betweenness centrality, reported in **Fig 7C**, show a dramatic difference in the extent of connectivity between the two domains which is mainly driven by hydrophobic contacts. In the three datasets, the [S:A222V + S:D614G] mutant forms at most a single strong "pivot" connection between the two domains (see violet community pairs 2$^{NTD}$-2$^{RBD}$ and 3$^{NTD}$-4$^{RBD}$ for UDD and *gly*-UDD respectively in **Fig 7C**), thereby allowing for a degree of flexibility in their relative geometry, while the S:D614G mutant forms 3–6 stable contacts (mainly involving communities 1 and 3 of the NTD with 2 and 4 of the RBD in **Fig 7C**). This last network is extensive enough to stabilize the RBD in a single dominant conformation. No stable connections were found between the RBD-NTD pair of the cryo-EM [S:A222V + S:D614G] mutant below the selected contact cut-off of 4.5 Å. As above, this trend shows up

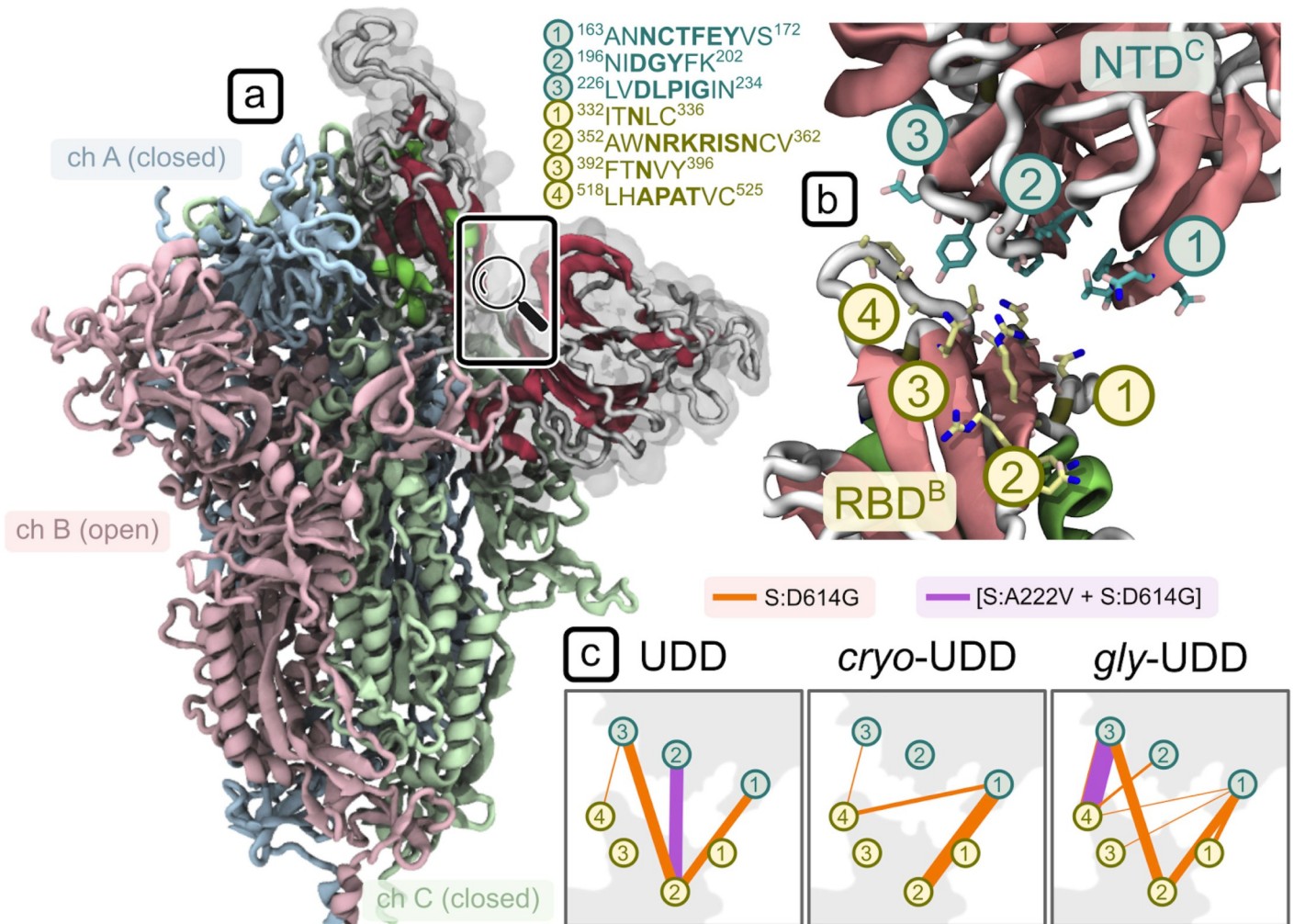

**Fig 7. Dynamic network analysis for a subsystem consisting of the open RBD and its neighbouring NTD. (a)** Open RBD of the simulated 1-up (UDD) SARS-CoV-2 Spike mutants. (**b**) Residue communities (3 for NTD, in teal, and 4 for RBD, in yellow) engaged in inter-domain contacts, along with their position in the protein sequence. The domains shown here are those highlighted with gray transparency in (a). (**c**) Communication pathways are highlighted with connecting lines whose thickness is proportional to the inter-domain betweenness centrality values. Orange connections represent the S:D614G and violet the [S:A222V + S: D614G] mutant; data is reported for the three systems: glycan-free from 6VSB (UDD), glycan-free from our cryo-EM data (*cryo*-UDD), and glycosylated from 6VSB (*gly*-UDD).

consistently across the three different setups—i.e., independent of both glycosylation and the source of the initial structure.

**Structural flexibility of PDB-reported S:D614G single mutant.**   In an attempt to provide more details about the dynamics of the S:D614G mutant, especially from a wider timescale perspective, a set of 24 structures of S:D614G previously reported in the PDB were also analysed by PCA together with our new cryo-EM structures. Note that the whole process of sample preparation for cryo-EM may take from seconds to minutes, therefore we expect that the individual images that support these PDB structures came from virtually any possible conformational state of the spike, although the maps themselves are the result of an image processing process which might not be able to accurately follow these structural variations. We also remark that for structures analysed in this PDB-wide approach, specimens may differ among themselves in more mutations than S:D614G and S:A222V alone, related to different ways to

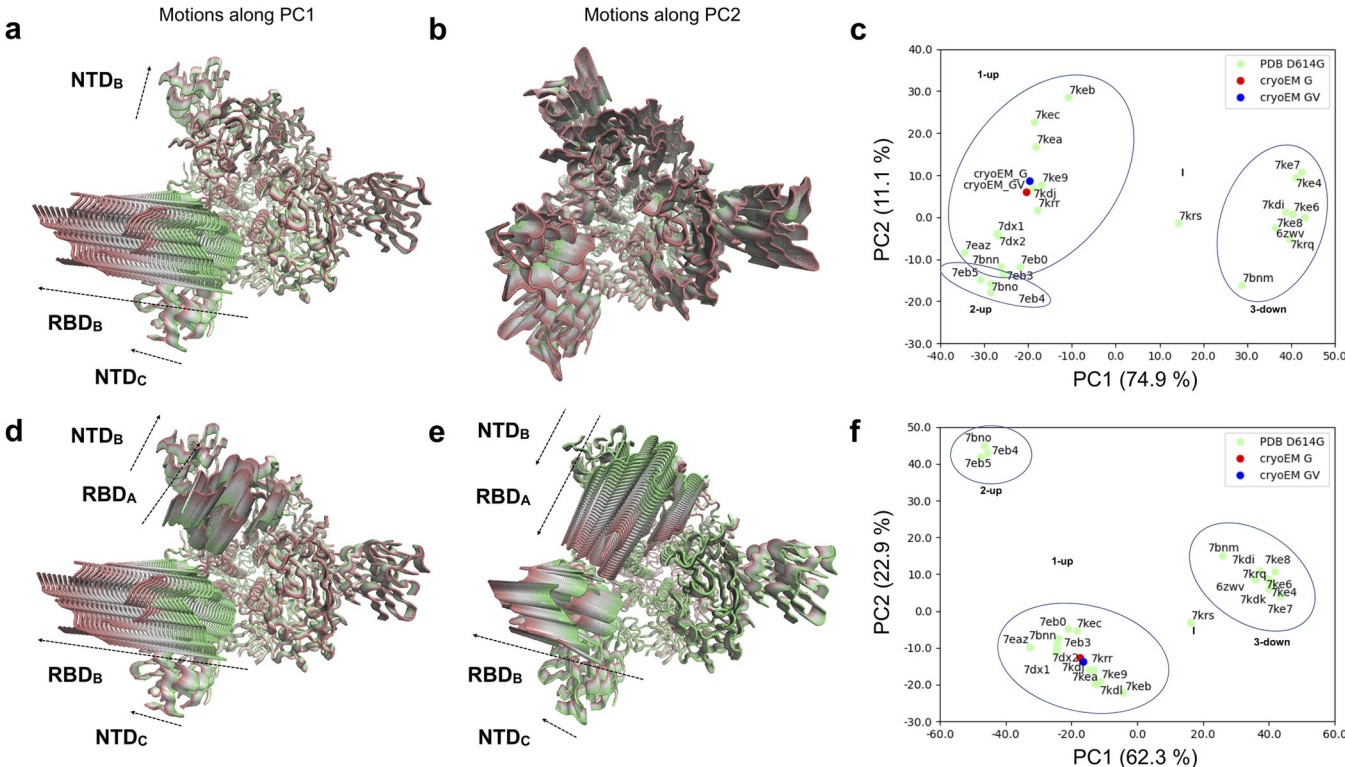

**Fig 8. PCA for PDB-reported S:D614G single mutant.** PCA results for the whole spike excluding **(a-c)** and including the 2-up conformations **(d-f)** in the covariance calculation. **(a)** Conformational evolution showing RBD opening motions observed in PC1 of the whole spike excluding 2-up conformations is shown from a top view ranging from a closed state (green) to an open state (red). **(b)** Conformational evolution along PC2 of the whole spike excluding 2-up conformations is shown from a top view from one extreme in green (compressed state) to the other in red (stretched state). Different domains move differently besides the compression and stretching of the structure. **(c)** Projection of the PDB S:D614G structures (green) and our cryo-EM structures (red and blue) onto the space of the first two PCs of the whole spike excluding 2-up conformations. Note a flipping of PC1 in this analysis relative to the PCAs from MD ensembles. **(d)** Concerted RBD opening motions observed in PC1 of the whole spike including 2-up conformations are shown from a top view ranging from one extreme in green (closed state) to the other in red (open state). **(e)** Anticorrelated RBD opening motions observed in PC2 of the whole spike including 2-up conformations are shown from a top view ranging from one extreme in green (closed state) to the other in red (open state). **(f)** Projection of the PDB S:D614G structures (green) and our 1-up cryo-EM structures (red and blue) onto the space of the first two PCs of the whole spike including 2-up conformations.

aid purification of stable spike trimers e.g. adding tags, eliminating cleavage sites and introducing proline mutations (**Table F in S1 Text**); still, all these structures were pooled together and analysed using PCA (**Fig 8** and **Figs L and M in S1 Text**).

The much smaller size of the ensemble as compared to that from MD simulations allowed us to analyse the dynamics of the whole spike, which produced very similar motions of the dominant subunit to those seen when focusing on the first S1 subunit involved in RBD opening as above, especially in the first three eigenvectors (**Fig L a-b in S1 Text**), while allowing us to identify motions of NTDs and RBDs from other subunits coupled to large motions of the dominant RBD (**Fig 8A** and **Fig L c in S1 Text**). However, to maintain a direct comparison with the MD data, we initially excluded the 2-up structures from the covariance calculation to avoid the extra component of variation associated with the second RBD opening, which has not been explored in our MD simulations.

In line with the PCA from MD simulations (**Fig 6**), the first eigenvector or principal component (PC1: 89.4%) from experimental S:D614G structures based on the single subunit is associated to the *closed-to-open* transition for the RBD together with a rearrangement of its NTD towards and away from the RBD. The equivalent motion of this subunit (chain B in our structures) in PC1 of the whole spike (overlap 0.99 and contribution 75.0%; **Fig L a-b in S1**

**Text**) also featured motions of the neighbouring NTD (chain C in our structures), which moved together with the dominant RBD (**Fig 8A**). The second eigenvector of both PCAs had a low contribution (4.9% and 10.9%) and was dominated by stretching and compression within it (**Fig L d in S1 Text**). This non-physiological variation in size of the structure is likely related to errors in cryo-EM data acquisition, processing and/or modelling. However, there are also more potentially interesting contributions of particular domains in this mode, including the second and third RBDs and the associated NTDs (**Fig 8B**), making it interesting to include.

As with the MD, projection of these 24 structures together with our new cryo-EM structures of the S:D614G and [S:A222V + S:D614G] mutants into the space of these two PCs (**Fig 8C**) allowed a separation of 3-down and 1-up structures along PC1, as well as an intermediate structure (PDB: 7KRS) [7] in this case. There were a number of structures with a greater degree of RBD opening relative to the two new cryo-EM structures, which occupied a similar place to each other in this landscape in line with their differences being related to more local motions, suggesting the possibility of greater opening in the S:D614G mutant too, albeit on longer time-scales than accessible to the MD simulations. Interestingly, there is also some separation along PC2 including separation of some 1-up structures including one with an intermediate conformation second RBD (upper region; PDB: 7KEC) [8] and 2-up structures (lower region; PDB: 7BNO, 7EB4 and 7EB5) [4,6] from other 1-up structures.

Finally, to better describe the full structural variation over the longer timescale captured by this set of PDB structures, we also performed an additional PCA by including the 2-up structures in the covariance calculation, allowing us to capture the associated variation in the second RBD as well (see **Fig M a in S1 Text**). In this case, PC1 and PC2 (62.3% and 22.9% contributions; **Fig M b in S1 Text**) both showed high overlaps with PC1 of the single subunit (1.00 and 0.86, respectively; **Fig M c in S1 Text**), suggesting that they were both dominated by opening and closing of the first RBD. Intriguingly, both PCs also featured significant motions of the second RBD (chain A of our structures), which is involved in the transition to 2-up (**Fig M d-e in S1 Text**). PC1 showed coupled opening and closing of these two RBDs (**Fig 8D**); whereas PC2 showed one RBD opening while the other closed (**Fig 8E**). There was a dominance of the first RBD in PC1 (**Fig M d in S1 Text**) and a dominance of the second RBD in PC2 (**Fig M e in S1 Text**), enabling a much clearer separation of 2-up structures in the projection onto these two PCs (**Fig 8F**). We again see that our two cryo-EM structures reside in the middle of the 1-up cluster, showing room for further opening of both dominant RBDs within the 1-up macrostate.

## Discussion

We started our study prompted by the emergence in Spain of variant 20E in summer 2020, deepening into this work as one of the key mutations characteristic of this variant, S:A222V, kept reappearing. Results from genomics, neutralization analysis, cryo-EM and MD were integrated to provide grounds to formulate a hypothesis answering some of the questions on SARS-CoV-2 evolution such as: Why do rare spike mutations keep reappearing in multiple backgrounds, including Delta, as is the case for S:A222V? It is worth mentioning that the epidemiological success of a mutation depends on the genetic background where it emerges but it can also just reflect underlying population genetic processes such as founder effects that impact the frequency of the mutation. S:A222V represents a good example of the aforementioned phenomenon. In fact, the success of 20E was likely driven by superspreading events and human mobility [10] rather than S:A222V *per se*. On the contrary, the case of AY.4.2 suggests a subtle increase in transmissibility of the virus that is likely associated with S:A222V and/or S:Y145H.

Here, we have carried out an in-depth evaluation of the impact of S:A222V with the aim to capture relevant features of the mutation. This multidisciplinary approach also offered the possibility to understand how *in silico* and *in vitro* protein structure analysis can be combined to predict the impact of SARS-CoV-2 spike mutations. In this way, the biophysical characterization of two types of samples (S:D614G and [S:A222V + S:D614G]) was carried out, including thermal stability assays and protein-protein interaction assays by biolayer interferometry. These studies showed that while S:A222V does not increase the stability of the spike protein, it slightly improves its affinity towards the host receptor, ACE2. This enhanced affinity is mainly driven by an increased association constant ($k_{on}$), in agreement with the increased flexibility of the RBD in the 1-up configuration suggested by both MD and cryo-EM studies.

Cryo-EM was used to obtain maps of the S:D614G and [S:A222V + S:D614G] mutants at global resolutions of 4.2 and 3.4 Å respectively from which structural models were derived. Images were processed using workflows oriented towards identifying symmetry breaking conditions characteristic of one or several of the RBDs in an up conformation. To avoid classification instabilities and inaccuracies in continuously flexible systems in which discrete flexibility classification methods were imposed [20], we performed a discrete classification analysis but enforcing results reproducibility through consensus [15]. Under these very stringent reproducibility conditions, we were able to report a majority population of 1-up structural classes and a minority population of 2-up structural classes in both S:D614G and [S:A222V + S:D614G], with similar percentages. We only found the 3-down conformation in a very small percentage of the [S:A222V + S:D614G] mutant images. However, just indicating the percentage of particles that had been reproducibly clustered together and then visually labelled as 1-up, 2-up or 3-down is a gross oversimplification of the informational content of cryo-EM maps. Furthermore, we might be close to the limits of what discrete 3D classification can obtain in a system like the one under study [20–22]. Here, a possible improvement could be provided by using new continuous flexibility approaches, whose practical limits still need to be tested [21–26]. Nevertheless, it is important to note that refinement of the cryo-EM structures in 1-up conformation of both the S:D614G and [S:A222V + S:D614G] mutants in their corresponding maps yielded B-factors that were notably different, especially at the RBD, suggesting that this latter domain was more flexible in the [S:A222V + S:D614G] mutant.

Molecular dynamics simulations were also carried out to provide complementary and perhaps less discrete structural details about the subject under study. Considering the importance of glycosylation for the spike dynamics [17–19], MD trajectories were collected starting from our cryo-EM structures as well as from previously reported structural models with and without glycosylation. Results from PCA relative to the RBD *closed-to-open* transition pointed out the preference for a more open RBD for the [S:A222V + S:D614G] mutant, with glycosylation additionally enhancing the openness of the open RBD, in agreement with a recent study [18]. Furthermore, dynamic network analysis allowed us to go deeper into the allosteric nature of the possible changes induced by the S:A222V mutation. These data support the hypothesis of a more pronounced uncoupling of the RBD from its neighbouring NTD for the [S:A222V + S:D614G] mutant when compared to the S:D614G mutant alone. This effect would be due to a lower conformational restraint imposed by the NTD on RBD dynamics in the 1-up conformation as a consequence of the reduced number of connections between these two domains in the [S:A222V + S:D614G] mutant (no special effects were observed on the 3-down conformation). Of note, this trend is consistently observed across all the simulated (glycan-free and glycosylated) systems. Interestingly, a mutational free energy analysis was not able to establish a thermodynamic preference for the open conformation between S:D614G and [S:A222V + S:D614G] mutants. This would suggest that although a wider RBD opening for the [S:A222V + S:D614G] mutant is possible it would not be associated with a significant energetic gain.

Finally, we compared all reported PDB structures of S:D614G (obtained by cryo-EM) as well as the ones presented here using Principal Component Analysis, and found similar motions to those seen by MD. Clearly the structures separate into 1-up, 2-up and 3-down (and an "intermediate" one), as expected, with the ones obtained in this work essentially in the middle of the 1-up cluster. Still, further work comparing MD simulations and cryo-EM maps needs to be performed, with continuous flexibility analysis being an essential component of these future analyses.

In conclusion, the S:A222V mutation at the NTD of the SARS-CoV-2 spike sequence is able to produce small but noticeable allosteric effects in the RBD that translate into changes in its conformational dynamics as well as biophysical properties. These findings agree with the recall of the S:A222V across waves, lineages, and space, suggesting a role for positive selection. Still, the genetic background in which the mutation occurs seems to be crucial for its success and the epidemiological effects of S:A222V mutation may depend on epistatic interactions between mutations, with S:L452R and S:T478K also potentially modifying the behaviour of the open RBD. Although predicting the combined outcome of different mutations on different backgrounds is still an open question, this work gets us a step closer by pinpointing the defined manner in which S:A222V affects key properties of the spike.

## Methods

### Sequence analysis and selection analysis

The global dataset of 4,993,996 sequences was obtained from GISAID database [27] since first case detected from 24 December 2019 until 1 April 2022 and filtering out sequences from non-human host, with no complete collection date, with low coverage or incomplete sequences (see **Data Availability**). 4,993,996 SARS-CoV-2 sequences were aligned with the Nextalign package (https://github.com/nextstrain/nextclade/tree/master/packages/nextalign_cli). Single nucleotide polymorphisms in amino acid position 222 of the spike were obtained after generating a VCF file from each individual fasta sequence derived from the alignment using SNP sites v 2.5.1 [28] with argument "-v" and using the reference genome (NC_045512.2) as the reference bases for detecting mutated sequences. B.1.617.2 sequences were obtained after selecting all sequences classified as B.1.617.2 PANGO lineage by GISAID from the global dataset described above. After eliminating sequences harbouring at least one indetermination (symbolized as "N"), and duplicated sequences detected with seqkit v 0.13.2 [29] (arguments employed: rmdup -s) we obtained a high quality B.1.617.2 dataset of 2,992,547 sequences. Finally, in order to reduce the number of sequences to a suitable dataset for phylogenetic reconstruction, we selected 10,049 sequences from the high-quality B.1.617.2 dataset randomly but keeping the same temporal distribution by month, the same proportion of B.1.617.2 sub-lineage and the same proportion of sequences with amino acid V in S:222 as the initial alignment of 2,992,547 sequences. To determine phylogenetic relationships of B.1.617.2 and the emergence of A222V, phylogenetic analysis was performed as previously described [30]. With the same methodology we performed a subset of sequences representing members of: B.1.177 PANGO lineage reducing 133,304 to 9,635 sequences, B.1.617.2 reducing 2,992,547 to 10,049 sequences, and G clade reducing the set of 4,993,996 sequences to 11,166 sequences.

In order to characterize codons under selective pressures in the NTD region of the spike (nucleotide coordinates: 21,677 to 24,298 in the reference genome), we calculated dN/dS (the ratio of the number of nonsynonymous substitutions per non-synonymous site to the number of synonymous substitutions per synonymous site) with phylogenetically corrected single-likelihood ancestor counting (SLAC) from Hyphy v 2.5.36 [13]. This ratio typically estimates the strength and direction of selection. Two approaches were conducted: first we analysed

sequences form the whole G-clade until April 2022 and second, we analysed four different pandemic periods. For the whole G clade, we reduced the total number of sequences (4,993,996 sequences) to two smaller datasets: 4,399 unique haplotypes and 13,459 unique haplotypes. The reduction of the complete dataset to a reduced number of haplotypes consisted in random selection of sequences with the same distribution of PANGO lineages and occurrence of S:A222V as the complete dataset, and then extracting unique haplotypes in Python. For the second approach we used four datasets, each one including a subset of sequences from the dates where each of four VOCs circulated. One consists of 2,673 sequences (1,124 haplotypes) from July 2020 to April 2021 that corresponds to B.1.177 circulation. The second with 4,758 sequences (1,972 haplotypes) from October 2020 to August 2021 corresponds to B.1.1.7 circulation. The third one includes 7,789 sequences (3,178 haplotypes) from March 2021 to March 2022 with the circulation of B.1.617.2. Lastly, the fourth dataset with 1,111 sequences (560 haplotypes) includes the period from December 2021 to March 2021, which includes the circulation of B.1.1.529.

## Plasmids

The plasmids used for the generation of the pseudotyped vesicular stomatitis virus (VSV) were previously described [30]. Plasmid pSPIKE (a generous gift from Cesar Santiago, CNB-CSIC) was designed to include the region encoding the SARS-CoV-2 protein S ectodomain (residues 15–1213) with substitutions to proline at residues 986 and 987 and of $^{668}$RRAR$^{671}$ furin cleavage site to alanine, with a N-terminal gp67 signal peptide for secretion, and C-terminal foldon trimerization motif, a thrombin protease recognition site and 9×His and Myc tags, into the insect expression plasmid pFastBac. Plasmid pACE2$_{TEV}$ was prepared from that previously used to express the N-terminal peptidase domain of human ACE2 [31], by including a cleavage site for TEV protease, to allow removing the C-terminal 6×His tag in an additional purification step.

## Evaluation of neutralization by convalescent-phase sera

Pseudotyped VSVDG-GFP bearing the ancestral Wuhan-Hu-1, the S:D614G, or 20E (mutations S:A222V and S:D614G) spike protein were produced as previously described [30]. Neutralization assays were performed as previously described [30]. Briefly, 16 h post infection GFP signal derived from VSV replication was determined in each well using an Incucyte S3 system (Essen Biosciences). The mean GFP signal observed in several mock-infected wells was subtracted from that of all infected wells, followed by standardization of the GFP signal in each well infected with antibody-treated virus to that of wells infected with mock-treated virus. Any low or negative values resulting from background subtraction were arbitrarily assigned a low, nonzero value (0.001). The serum dilutions were then converted to their reciprocal and the dose resulting in a 50% reduction in GFP signal was calculated using the R drc package v 3.0–1. A three-parameter log-logistic regression (LL3 function) was used for all samples. All serum samples were from donors that were admitted to the intensive care unit during April 2020 (ethics approval RVB20017COVID).

## Site directed mutagenesis and protein production

Site-directed mutagenesis of pSPIKE was performed using Q5 Site-Directed Mutagenesis Kit (New England Biolabs) and the forward and reverse primers given in **Table G in S1 Text**. The correctness of the constructs, the presence of the desired mutation, and the absence of unwanted mutations were corroborated by sequencing.

For producing SARS-CoV-2 protein S variants (Wuhan-Hu-1, S:D614G or [S:A222V + S:D614G]) we used the Bac-to-Bac Baculovirus Expression System (Invitrogen). *E. coli* DH10Bac cells (Invitrogen), transformed with the appropriate pSPIKE construct carrying either Wuhan-Hu-1, S:D614G or [S:A222V + S:D614G] variants, were grown on LB-agar containing 50, 7, 10, 40 and 100 µg/ml of, respectively, kanamycin, gentamycin, tetracyclin, IPTG and Bluo-Gal. Individual white colonies were inoculated into 5-ml LB medium with the same antibiotics, cultured overnight (37˚C, orbital shaking at 180 rpm), and the bacmid produced was isolated. The baculovirus was produced by transfecting 1 ml of Sf9 insect cells ($0.9 \times 10^6$ cells/ml in Grace medium, Invitrogen) with 45 µg of recombinant bacmid carrying the DNA-encoding sequences for Wuhan-Hu-1, S:D614G or [S:A222V + S:D614G] variants (proven by PCR), using 0.65% Fugene (Promega). After 5 hr of plate incubation at 27˚C, Grace medium was replaced by Sf900 medium (Invitrogen) containing 0.1% Pluronic F-68, 50 U/ml penicillin and 50 µg/ml streptomycin and cells were cultured for 4 days at 27˚C. At the end of the 4 days, the culture medium containing low baculoviral titres, was collected by centrifugation and used to infect a suspension of $1.5 \times 10^6$ Sf9 cells/ml by diluting it 60 fold in the cell suspension. After 96-hr culturing (27˚C, orbital shaking at 125 rpm) and centrifugation, the high-titre baculovirus supernatant produced was collected and used to infect Sf9 cells at a density of $3 \times 10^6$ Sf9 cells/ml by 1:8 dilution of the supernatant virus stock. After 4 days of culture, the produced variant of SARS-CoV-2 spike protein was purified from the medium, where the protein was secreted, by spinning out the cells (1 hour, $19,000 \times g$) and bringing the pH to neutrality by addition of 10% volume of a solution containing 0.5 M Tris-HCl pH 7.4, 60 mM KCl, 2.8 mM NaCl.

Subsequent steps were carried out at 4˚C unless indicated. A peristaltic pump was used to apply the culture supernatant (700–1000 mL) to a 5 mL-HisTrap Excel column (Cytiva) equilibrated with 20 mM Na-Hepes pH 7.2, 150 mM NaCl (buffer A). After mounting the column to al FPLC, and washing with buffer A supplemented with 25 mM imidazole, SARS-CoV-2 S protein was eluted with 0.5 M imidazole in buffer A, collecting fractions of 2 ml. Fractions containing SARS-CoV-2 S protein (shown by SDS-PAGE) were pooled and concentrated to 0.5 ml by centrifugal ultrafiltration (100-kDa cutoff membrane, Amicon Ultra; Ultracel). The SARS-CoV-2 S protein was then purified by size exclusion chromatography (SEC) using a Superdex 200 Increase 10/300 GL column (Cytiva) pre-equilibrated with 10 mM sodium-Hepes pH 7.2, 150 mM NaCl (SEC buffer), collecting 0.2-ml fractions, of which those containing SARS-CoV-2 S protein were aliquoted, flash-frozen in liquid nitrogen and stored at -80˚C. SARS-CoV-2 S protein concentration was determined spectrophotometrically from the optical absorption at 280 nm, using a sequence-deduced (EXPASY ProtParam tool, https://www.expasy.org) mass extinction coefficient (E 1%) of 10.29 $g^{-1}L$ $cm^{-1}$.

ACE2 was produced and purified essentially as described for SARS-CoV-2 protein S, except for the fact that before SEC, the polyhistidine tag was removed by digestion with TEV protease (37.5 µg per each mg of ACE2, overnight, 4˚C) within a dialysis bag dialyzed against SEC buffer supplemented with 5 mM EDTA and 0.5 mM mercaptoethanol. Nontagged ACE2 was isolated by an additional Ni-affinity chromatography step, using a 1 mL-HisTrap Excel column (Cytiva). In this step, the untagged protein was collected in the initial fraction not retained by the column (application and 2-mL wash). After concentration by centrifugal ultrafiltration (30-kDa cutoff membrane), the protein was further purified by SEC as described for SARS-CoV-2 protein S, quantifying ACE2 by its optical absorption at 280 nm (mass extinction coefficient, 21.9 $g^{-1}$ L $cm^{-1}$ was considered as mass extinction coefficient; E1%). The absence of the polyhistidine tag in the purified protein was verified by western blot using antihistidine antibodies.

## In vitro functional assays

**Thermal shift assays.** Thermofluor assays [32] were performed in 20 μl of a solution of 0.1 mg/ml SARS-CoV-2 spike protein (Wuhan-Hu-1, S:D614G, or [S:A222V + S:D614G] variants), in 10 mM Na Hepes pH 7.3, 150 mM NaCl and a 1:1,000 dilution of SYPRO Orange (commercial preparation from Invitrogen, Carlsbad, CA), kept in sealed microwell plates. A real-time PCR instrument (7500 model from Applied Biosystems, Thermo Fisher Scientific, Alcobendas, Madrid, Spain) was used to monitor the increase in SYPRO Orange fluorescence (excitation at 488 nm; emission at 610 nm) with temperature increase at a ramp of 1˚C/min. The samples were preincubated 20 minutes at 37˚C prior to performing the assays [33]. Plots, curve fittings and numerical calculations were performed with the program GraphPad Prism 5 (GraphPad Software, San Diego, CA, USA). Results given in Table 1 correspond to the mean and standard error for six replicate assays carried in groups of three replicates in two independent experiments.

**Biolayer interferometry assays.** Binding assays of ACE2 to His-tagged S proteins were performed in the Octet K2 instrument (ForteBio). The assays were carried out at 28˚C, shaking was kept at 1,000 rpm and the solution was 20 mM Hepes, 150 mM NaCl, 50 μM EDTA, 10 mM imidazole and 0.005% Tween 20 (buffer AB). His-tagged Spike proteins at 0.1 μg/ml in solution AB were immobilized (240 s) on Ni-NTA biosensors hydrated in the same buffer, yielding a typical signal of ~2 nm. After another 240 s equilibration of the biosensors with buffer AB to get a stable baseline, binding was assayed by adding serial 1:2 dilutions of ACE2 in the range 250–3.9 nM. Both the association and the dissociation phases lasted 120 s each. Negative drift controls were carried out without ACE2 and a 1:1 binding model was used for data fitting.

## Cryo-electron microscopy sample preparation

**Cryo-EM data acquisition.** Purified SARS-CoV-2 spike samples in 10mM Hepes pH7.2 and 150mM NaCl (3 μL at 0.62–0.75mg/ml) were kept at 4˚C until their application onto QUANTIFOIL R 1.2/1.3 Cu:300-mesh grids (QUANTIFOIL). The grids were vitrified using a Leica GP automatic vitrification robot (Leica). Chamber conditions were set at 10˚C and 95% relative humidity. Grids were glow discharged for 30 seconds prior to application of the samples. Data were collected on a FEI Talos electron microscope operated at 200 kV and images recorded on a FEI Falcon III detector operating in electron counting mode. A total of 4,841 and 2,492 movies were recorded for [S:A222V + S:D614G] and S:D614G, respectively; at a calibrated magnification of 120,000x, yielding a pixel size of 0.85 Å on the specimen. Each movie comprises 60 frames with an exposure rate of 0.54 e-/Å$^2$ per frame, with a total exposure time of 20 s and an accumulated exposure of 32.4 e-/Å$^2$. Data acquisition was performed with EPU Automated Data Acquisition Software for Single Particle Analysis (Thermo Fisher Scientific) at -0.3 μm to -3.5 μm defocus.

**Image processing.** All image processing steps were performed inside Scipion [34]. Movies were motion-corrected and dose weighted with MOTIONCOR2 [35]. Aligned, non-dose-weighted micrographs were then used to estimate the contrast transfer function (CTF) with GCTF [36] and CTFFIND4.1 [37]. The Scipion CTF consensus protocol [15] was used to select 4,330 and 1,831 micrographs for [S:A222V + S:D614G] and S:D614G, respectively (**Fig G in S1 Text** and **8**). Micrographs were then automatically picked using gautomatch and crYOLO [38]. Following the application of the Scipion picking consensus protocol 509,466 and 268,763 particles were extracted for [S:A222V + S:D614G] and S:D614G, respectively. From now on pairs of values will be given for each step in the order corresponding to these two variants of S. Images were binned to 1.40 Å/px from this step onwards. 2D classification was performed in

cryoSPARC [39] and 310,162 and 165,304 particles were selected. The CryoSPARC initial model protocol was then used to generate and classify the particles into 4 classes, with and without imposing C3 symmetry. Several datasets being more or less restrictive with the number of classes and thus particles selected, were refined using non-uniform refinement in cryoS-PARC with no symmetry application, to resolutions of 3.4 Å and 4.2 Å based on the gold-standard (FSC = 0.143) criterion. Several datasets were also selected for applying the C3 symmetry approach, being refined to resolutions of 3.2 Å and 3.7 Å. The dataset with the highest number of particles was then 3D classified in RELION 3.1 [40] using the symmetry relaxation protocol [41–42], which relaxes the dominant symmetry by considering multimodal priors centered at the symmetry related positions. A 30 Å low-pass filtered reconstruction with no symmetry resulting from a randomly selected subset of 10,000 symmetry broken particles [42] was used as the initial model for the 3D classification. This protocol was repeated three times with the same parameters and to overcome the variability in the results, the Scipion 3D classification consensus protocol was used to better characterize the different conformations [15]. An initial model was generated for each class/conformation and subsequently refined to a resolution of 3.3–3.4 Å ([S:A222V + S:D614G]-1-up), 6.6–6.8 Å ([S:A222V + S:D614G]-2-up), 7.0 Å ([S:A222V + S:D614G]-3-down), 4.1–4.2 Å (S:D614G-1-up) and 8.0–8.8 Å (S:D614G-2-up). The resulting maps were sharpened with DeepEMhancer [43].

**Model building and refinement.** For model building of [S:A222V + S:D614G] in the 1-up conformation ([S:A222V + S:D614G]-1-up), we started with a deposited PDB file for the S protein in the 1-up conformation (PDB: 7BNN). After a simple fitting of 7BNN to the [S:A222V + S:D614G]-1-up map using Chimera [44], a rigid-body docking of different regions of each monomer for keeping the secondary structure of Spike (14–293; 294–319 plus 592–699; 320–330; 331–529 plus 530–591; 734–775; 944–962; 963–989; 990–1027) was done using Coot. Manual building, deletion of disordered loops in the model as well as deletion and/or addition of glycosylations were done in Coot followed by several rounds of refinement in REFMAC [45]. This process was repeated until acceptable refinement metrics were obtained (see **Table B in S1 Text**). The refined [S:A222V + S:D614G]-1-up model was used as the starting point for model building of S:D614G-1-up. Again, a simple fitting with Chimera using our [S:A222V + S:D614G]-1-up model and the S:D614G-1-up map was done, followed by a rigid-body docking of the same regions as described above. After several rounds of refinement in REFMAC and model building in Coot [46], acceptable refinement metrics were obtained (see **Table B in S1 Text**). In both models, the lower resolution in a few regions made it difficult to define the structures for some loops (residues 621–641 and 676–689 for S1 and residues 827–854 for S2). Thus, all these procedures generated two models, one for S:D614G-1-up and another one for [S:A222V + S:D614G]-1-up, and both were used for the structural studies performed in this work.

## Molecular dynamics simulations

**System preparation.** The closed, 3-down and open, 1-up states of the glycan-free ectodomain (head-only) of the SARS-CoV-2 Spike glycoprotein were retrieved from the COVID-19 archive (https://www.charmm-gui.org/?doc=archive&lib=covid19) and used to generate the closed and open states for the 20E (EU1) mutant. The two models were built respectively from the cryo-EM structure of the wild type Spike glycoprotein in its 3-down (named "DDD"; PDB: 6VXX) [47] and 1-up (named "UDD"; PDB: 6VSB) [48] conformations. Point mutations at position 222 (A—>V) of the S1-NTD domain (residues 13–305) and 614 (D—>G) of the S1/S2 furin cleavage site were introduced by using the mutagenesis tool of PyMol 2.0 (glycan-free) or an in-house topology editing tool (glycosylated; see gitlab.com/KomBioMol/

gromologist). For comparative purposes, the S:D614G mutant was also simulated. Accordingly, four systems named DDD$_{S:D614G}$, DDD$_{[S:A222V + S:D614G]}$, UDD$_{S:D614G}$ and UDD$_{[S:A222V + S:D614G]}$ and another two models named *gly*-UDD$_{S:D614G}$ and *gly*-UDD$_{[S:A222V + S:D614G]}$ for the glycosylated spike were examined. The same protocol was applied to prepare the cryo-EM (glycan-free) structures, *cryo*-UDD$_{S:D614G}$ and *cryo*-UDD$_{[S:A222V + S:D614G]}$ which were also subjected to MD simulations. The Amber ff99sb-ildn force field [49–50] was used to simulate the protein. Standard protonation states at pH 7.4 were adopted for ionizable residues. The same N- and O-glycosylation profile, consisting of 22 N-glycans and 1 O-glycan per each monomer as described in the work of Woo H. et al. 2020 [51] was applied for the simulation. Finally, for all the simulated systems, the protein was enclosed in a truncated octahedron box, solvated with TIP3P [52] water molecules and neutralized by adding Na$^+$ or K$^+$/Cl$^-$ counterions [53]. A detailed description of all the MD systems studied in this work is reported in **Table D in S1 Text**.

**MD simulations.** Classical MD simulations were carried out with Gromacs v. 2020 [54]. The systems were energy minimized by applying 50,000 steps of the steepest descent algorithm followed by 5,000 steps of conjugate gradient algorithm. 1 ns of MD simulation in the NVT ensemble [55] was run using the velocity-rescaling thermostat and a time coupling constant of 0.1 ps to heat the systems to the production temperature of 300 K. Positional restraints with a force constant of 1,000 kJ mol$^{-1}$ mn$^{-2}$ were applied to the protein backbone atoms to avoid unnatural distortions during heating. Equilibration to the target temperature was then accomplished in two steps consisting of (i) 5 ns of unrestrained equilibration in the NPT ensemble using Berendsen thermostat (0.1 ps time coupling constant) and barostat (0.5 ps time coupling constant) and (ii) 5 ns of unrestrained MD simulation in the NPT ensemble by using Berendsen thermostat (0.1 ps time coupling constant) and Parrinello-Rahman barostat (0.5 ps time coupling constant) [56]. For the glycan-free systems, 300 ns of NPT MD production were run for the 3-down (DDD) state in periodic boundary conditions by using the Parrinello–Rahman barostat. For the 1-up (UDD) state, a longer MD run (500 ns) was performed to enhance conformational sampling. All the simulations were run in triplicate, leading to a global simulation time of 2.4 μs (0.9 + 1.5 μs; **Table D in S1 Text**) for the S:D614G and [S:A222V + S:D614G] mutants. For the fully-glycosylated systems based on 6VSB and the cryo-EM (glycan-free) systems, respectively, 400 and 200 ns of NPT MD production were run for the UDD models of both the S:D614G and [S:A222V + S:D614G] mutants, leading to a global simulation time of 1.2 μs (0.8 + 0.4 μs; **Table D in S1 Text**). In all the cases, the LINCS method [57] was applied to constraint bonds involving hydrogen atoms. A cut-off of 1.2 nm was used to treat short-range nonbonded interactions, whereas the PME method was applied to manage long-range electrostatic interactions [58]. A time step of 2 fs was applied to collect trajectories during the simulation.

**Convergence assessment.** Root-mean-square deviation (RMSD) analysis was carried out to evaluate the structural stability and convergence of the simulated systems. The *gmx rms* tool of gromacs 2020 was used to calculate the 1D positional rmsd for the protein Cα atoms over time. The RMSD profiles (**Fig N in S1 Text**) supported the structural stability for each of the three replicas for the S:D614G and [S:A222V + S:D614G] mutants in the closed conformation (**Fig N a-b in S1 Text**), as noted in average RMSD values ranging between 0.2 and 0.4 nm with respect to the starting, energy minimized structure. Slightly higher deviations (up to 0.7 nm) were observed for the open state (**Fig N c-d in S1 Text**) although all the simulated systems globally reached structural convergence after 150 ns of MD simulation. The same analysis was done on the MD trajectories for the mutants (i) based on 6VSB and on (ii) the cryo-EM (glycan-free) structures and reported in **Fig N e-f in S1 Text**.

**Principal components analysis (PCA).** The internal collective motions of the principal S1 subunit for the S:D614G and [S:A222V + S:D614G] mutants of the Spike protein were analysed by calculating the positional covariance matrix for the Cα backbone atoms. For this analysis, all trajectories were re-aligned on the protein Cα atoms of the central helixes of the trimeric S2 domains (Cα atoms for residues $718_A$-$994_A$, $718_B$-$994_B$ and $718_C$-$994_C$). For each simulated glycan-free system (S:D614G and [S:A222V + S:D614G]), all frames across all replicas for the concatenated DDD and UDD trajectories were considered. The gromacs tool *gmx covar* was used to extract the eigenvalues and eigenvectors from the 2.4 and 0.4 μs-long MD trajectories, and *gmx anaeig* was used to analyse and plot the eigenvectors. For comparative purposes, the [S:A222V + S:D614G] mutant was always projected on the eigenvectors of the reference S:D614G mutant. For consistency, MD trajectories collected from cryo-EM and glycosylated mutants were also aligned on the previously cited ensemble. Another PCA focused on residues pertaining to the three RBD (residues $330_A$-$530_A$, $330_B$-$530_B$, $330_C$-$530_C$) was also performed and used to calculate continuous density distributions along the first eigenvector ($PC1^{RBD}$). Results from this analysis were compared with those obtained for projection along the first eigenvector ($PC1^{S1}$) for the whole S1 (residues $30_A$-$650_A$, $30_B$-$650_B$, $30_C$-$650_C$).

**Dynamic network analysis.** The dynamic network analysis was performed using an implementation by Melo *et al.* [59], following a standard set of parameters, i.e. a contact cut-off of 4.5 Å and contact persistence of 0.5. Since each system consisted of the same selection (open RBD + the neighbouring NTD), the reported values of betweenness centrality are absolute rather than normalized within a system. ParmEd [60] was used to convert between trajectory and topology formats as required for the analysis.

**Mutational free energy analysis.** Free energy calculations were performed using the non-equilibrium alchemical protocol based on the Crooks theorem [61]. The PMX library was used to introduce the modifications [62], and an in-house script (gitlab.com/KomBioMol/crooks) was used to run and analyse the free energy calculations. Bennett acceptance ratio (BAR) was used to determine the free energy values [63–64]. Starting points for individual non-equilibrium runs were sampled from 200-ns equilibrium simulations of the respective Spike mutant in the 1-up conformation (prepared as described above), and calculations were repeated for each of the three chains to compare the effect of mutations on the thermodynamic preference for the open, 1-up state.

**PCA of existing S:D614G structures.** 24 structures from six recent structural studies of the S:D614G spike were analysed together with our two new cryo-EM structures (**Table F in S1 Text**). More structural models were available in the PDB but were not included as they came from earlier rounds of classification as structures that were included (PDB: 7KDK and 7KDL) [8] or had a poorly resolved S1 subunit (PDB: 6XS6) [5]. Principal component analysis was performed using the version 2.0 of the Protein Dynamics (*ProDy*) application programming interface (API) [65] from GitHub as follows. Only Cα atoms were included for efficiency. A structural ensemble was created using a WT 1-up structure (PDB: 6VSB) as a reference for initial alignment, where the chains were relabelled to match our cryo-EM structures. Residues were matched with the default method and chains with a custom matching function using the chain order correspondences in the second last column of **Table F in S1 Text**, based on a visual analysis in PyMOL. Initial alignment and superposition to the reference was then followed by iterative superposition until the mean structure converged. According to the protocol used for the PCA of our MD simulations, structures were then iteratively superposed again using only the Cα atoms of core helices of the S2 trimer. The ensemble was then trimmed to exclude residues only found in some structures. Positional covariance matrices were calculated based on the single subunit related to RBD opening as done for the MD simulations as well as the whole spike. In the case of the whole spike, covariance matrices were calculated using both

the whole ensemble and a subset lacking the three 2-up structures. The three resulting covariance matrices were each decomposed to yield the first 20 eigenvectors describing the most significant modes of structural variation with eigenvalues corresponding to the extent of variance covered. One to three modes, which contributed >90% of the variance, were considered the principal components (PCs). Square fluctuations were calculated from all modes as eigenvalue-weighted dot products of corresponding eigenvectors. Matrices of directional overlaps were calculated as correlation cosines. For comparison with the single subunit PCA, the eigenvectors from the whole spike were sliced to only include elements corresponding to the single subunit. Conformations along individual eigenvectors were generated using the Normal Mode Wizard (NMWiz) [66] in VMD [67] and 40 conformers with an RMSD up to 4 Å in both directions were used for making Figs.

## Supporting information

**S1 Text.** Figs A-N and Tables A-G. The structural role of SARS-CoV-2 genetic background in the emergence and success of spike mutations: the case of the spike A222V mutation (DOCX)

**S1 Movie. Structural changes from S:D614G to [S:A222V + S:D614G], starting from cryo-EM maps, in two different orientations.** The subtle rearrangement of the RBDs and NTDs of subunits A and B of the spike is specifically highlighted. Position of residue 222 in each subunit is represented as a cyan sphere. (MP4)

**S1 Table. Full Table of frequencies of sequences with A222V for the different PANGO lineages.** (XLSX)

## Acknowledgments

We want to particularly acknowledge the patients and the Consorcio Hospital General de Valencia Biobank integrated in the Valencian Biobanking Network for their collaboration in providing the convalescent serum samples. We acknowledge access to the cryoEM CNB-CSIC facility in the context of the CRIOMECORR project (ESFRI-2019-01-CSIC-16), and in particular the help of its staff. CPU time was partially provided by the PL-Grid Infrastructure.

### Consortia

### The IBV-Covid19-Pipeline

Laura Villamayor (orcid.org/0000-0002-7654-3306), Carolina Espinosa, Anmol Adhav (orcid.org/0000-0002-3504-8675), Maria del Pilar Hernández-Sierra, Rafael Ruiz-Partida (orcid.org/0000-0002-1696-6753), Jesus Rodríguez-Díaz (0000-0002-9698-7684).

## Author Contributions

**Conceptualization:** Ron Geller, Iñaki Comas, Carmen Gil, Mireia Coscolla, Modesto Orozco, José Luis Llácer, Jose-Maria Carazo.

**Data curation:** Clara Marco-Marín, Carlos P. Mata, Paula Ruiz-Rodriguez, Maria Luisa López-Redondo, Roberto Melero, Carlos Óscar Sánchez-Sorzano, Marta Martínez, Rocío Arranz, José Luis Llácer.

**Formal analysis:** Tiziana Ginex, Clara Marco-Marín, Miłosz Wieczór, Carlos P. Mata, James Krieger, Paula Ruiz-Rodriguez, Maria Luisa López-Redondo, Clara Francés-Gómez, Roberto Melero, Carlos Óscar Sánchez-Sorzano, Marta Martínez, Alicia Forcada-Nadal, Roberto Gozalbo-Rovira, Rocío Arranz, José Luis Llácer.

**Funding acquisition:** James Krieger, Vicente Rubio, Alberto Marina, Ron Geller, Iñaki Comas, Mireia Coscolla, José Luis Llácer, Jose-Maria Carazo.

**Investigation:** Tiziana Ginex, Clara Marco-Marín, Miłosz Wieczór, James Krieger, Paula Ruiz-Rodriguez, Maria Luisa López-Redondo, Clara Francés-Gómez, Roberto Melero, Nadine Gougeard, Alicia Forcada-Nadal, Sara Zamora-Caballero, Roberto Gozalbo-Rovira, Carla Sanz-Frasquet, Rocío Arranz.

**Methodology:** Tiziana Ginex, Clara Marco-Marín, Miłosz Wieczór, Carlos P. Mata, James Krieger, Maria Luisa López-Redondo, Clara Francés-Gómez, Nadine Gougeard, Rocío Arranz, Jose-Maria Carazo.

**Project administration:** Jose-Maria Carazo.

**Resources:** Jeronimo Bravo, Vicente Rubio, Alberto Marina, Iñaki Comas, Mireia Coscolla, José Luis Llácer, Jose-Maria Carazo.

**Software:** Miłosz Wieczór, James Krieger, Paula Ruiz-Rodriguez.

**Supervision:** Clara Marco-Marín, Jeronimo Bravo, Vicente Rubio, Alberto Marina, Ron Geller, Iñaki Comas, Carmen Gil, Mireia Coscolla, Modesto Orozco, José Luis Llácer, Jose-Maria Carazo.

**Visualization:** Tiziana Ginex, Clara Marco-Marín, Miłosz Wieczór, Carlos P. Mata, James Krieger, Paula Ruiz-Rodriguez, Maria Luisa López-Redondo, Clara Francés-Gómez, Alicia Forcada-Nadal, Roberto Gozalbo-Rovira, Alberto Marina, José Luis Llácer.

**Writing – original draft:** Tiziana Ginex, Clara Marco-Marín, Miłosz Wieczór, Carlos P. Mata, James Krieger, Paula Ruiz-Rodriguez, Maria Luisa López-Redondo, Alicia Forcada-Nadal, Jeronimo Bravo, Vicente Rubio, Alberto Marina, Ron Geller, Iñaki Comas, Carmen Gil, Mireia Coscolla, Modesto Orozco, José Luis Llácer, Jose-Maria Carazo.

**Writing – review & editing:** Tiziana Ginex, Clara Marco-Marín, Miłosz Wieczór, James Krieger, Paula Ruiz-Rodriguez, Vicente Rubio, Mireia Coscolla, José Luis Llácer, Jose-Maria Carazo.

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
