## [Decision Letter · Decision Letter 0]

31 Mar 2022

Dear Prof. Carazo,

Thank you very much for submitting your manuscript "The role of SARS-CoV-2 genetic background in the emergence and success of spike mutations: the case of the spike A222V mutation" for consideration at PLOS Pathogens. As with all papers reviewed by the journal, your manuscript was reviewed by members of the editorial board and by several independent reviewers. The reviewers appreciated the attention to an important topic. Based on the reviews, we are likely to accept this manuscript for publication, providing that you modify the manuscript according to the review recommendations.

Sincerely,

Mark T. Heise

Section Editor

PLOS Pathogens

Mark Heise

Section Editor

PLOS Pathogens

Kasturi Haldar

Editor-in-Chief

PLOS Pathogens

orcid.org/0000-0001-5065-158X

Michael Malim

Editor-in-Chief

PLOS Pathogens

orcid.org/0000-0002-7699-2064

Reviewer Comments (if any, and for reference):

Reviewer's Responses to Questions

**Part I - Summary**

Reviewer #1: This study described the S:A222V point mutation and its impact of such mutation. The authors performed serological, functional, structural and computational studies combined to reveal the circulation of such mutation and suggesting it doesn't affect vaccine effectiveness.

Reviewer #2: This manuscript from Ginex and colleagues characterizes the effect of the A222V substitution on the SARS-CoV-2 spike protein. This substitution has been selected multiple times in the evolution of SARS-CoV-2, suggesting that it provides an advantage to the virus. The investigators show that A222V does not provide escape from antibody-mediated neutralization, suggesting that it does not affect vaccine efficacy. Cryo-EM structures of an A222V/D614G spike protein and a D614G control spike protein reveal that the two structures are similar, as expected for a single amino acid substitution. MD simulations reveal that the A222V spike has a higher propensity to sample the open RBD state, which would improve attachment of the virus to the host cells.

The biophysical studies are performed to a high level, and the analyses are thorough. The cryo-EM studies are described well but I didn’t see a table for the cryo-EM model validation statistics, which needs to be included. As I am not an expert on MD simulations, I cannot comment on this aspect of the manuscript. Given that the MD results are key to supporting the major claim of the manuscript (that A222V increases RBD opening), it is difficult for this reviewer to determine whether the conclusions are supported by the data. The conclusion appears plausible, and would provide an advantage to the virus, supporting the repeated selection of A222V.

**Part II – Major Issues: Key Experiments Required for Acceptance**

Reviewer #1: 1. The authors should define such mutation in the scope of lineage/sub-lineage/sub-clade. More importantly, a clear and detailed phylogenetic analysis (not that in Fig 1c) including these mutants within G clade should be provided.

2. The fitness related analysis for such mutants should be provided at least to show the natural selection of them. If the sequences are too many to be analyzed, I suggest to add some discussion for it.

Reviewer #2: (No Response)

**Part III – Minor Issues: Editorial and Data Presentation Modifications**

Reviewer #1: (No Response)

Reviewer #2: A table containing the cryo-EM model validation statistics needs to be provided. The quality of the models, which were deposited in the Protein Data Bank with codes 7QDG and 7QDH, cannot be assessed without the table or PDB validation reports.

PLOS authors have the option to publish the peer review history of their article (what does this mean?). If published, this will include your full peer review and any attached files.

Reviewer #1: No

Reviewer #2: No

Figure Files:

Data Requirements:

Reproducibility:

References:

---

## [Editor Report · Decision Letter 1]

1 Jun 2022

Dear Prof. Carazo,

We are pleased to inform you that your manuscript 'The structural role of SARS-CoV-2 genetic background in the emergence and success of spike mutations: the case of the spike A222V mutation' has been provisionally accepted for publication in PLOS Pathogens.

Best regards,

Mark T. Heise

Section Editor

PLOS Pathogens

Mark Heise

Section Editor

PLOS Pathogens

Kasturi Haldar

Editor-in-Chief

PLOS Pathogens

orcid.org/0000-0001-5065-158X

Michael Malim

Editor-in-Chief

PLOS Pathogens

orcid.org/0000-0002-7699-2064
---

## [Editor Report · Acceptance letter]

4 Jul 2022

Dear Prof. Carazo,

We are delighted to inform you that your manuscript, "The structural role of SARS-CoV-2 genetic background in the emergence and success of spike mutations: the case of the spike A222V mutation," has been formally accepted for publication in PLOS Pathogens.

Best regards,

Kasturi Haldar

Editor-in-Chief

PLOS Pathogens

orcid.org/0000-0001-5065-158X

Michael Malim

Editor-in-Chief

PLOS Pathogens

orcid.org/0000-0002-7699-2064